# Understanding and Enhancing Message Passing on Heterophilic Graphs via Compatibility Matrix

**Zhuonan Zheng**[★♠], **Yuanchen Bei**[♠], **Zhiyao Zhou**[♠], **Sheng Zhou**[★‡∗],
**Yao Ma**[†], **Ming Gu**[★♠], **Hongjia Xu**[★♠], **Jiawei Chen**[♠], **Jiajun Bu**[★♠]
★ Zhejiang Key Laboratory of Accessible Perception and Intelligent Systems, Zhejiang University
♠College of Computer Science and Technology, Zhejiang University
‡School of Software Technology, Zhejiang University
† School of Science, Rensselaer Polytechnic Institute

## Abstract

Graph Neural Networks (GNNs) excel in graph mining tasks thanks to their message-passing mechanism, which aligns with the homophily assumption. However, connected nodes can also exhibit inconsistent behaviors, termed *heterophilic* patterns, sparking interest in heterophilic GNNs (HTGNNs). Although the message-passing mechanism seems unsuitable for heterophilic graphs owing to the propagation of dissimilar messages, it is still popular in HTGNNs and consistently achieves notable success. Some efforts have investigated such an interesting phenomenon, but are limited in the *data* perspective. The *model*-perspective understanding remains largely unexplored, which is conducive to guiding the designs of HTGNNs. To fill this gap, we build the connection between node discriminability and the compatibility matrix (CM). We reveal that the effectiveness of the message passing in HTGNNs may be credited to increasing the proposed *Compatibility Matrix Discriminability (CMD)*. However, the issues of sparsity and noise pose great challenges to leveraging CM. Thus, we propose CMGNN, a novel approach to alleviate these issues while enhancing the CM and node embeddings explicitly. A thorough evaluation involving 13 datasets and comparison against 20 well-established baselines highlights the superiority of CMGNN.

## 1 Introduction

Graph Neural Networks (GNNs) have shown remarkable performance in graph mining tasks, such as social network analysis [1, 2] and recommender systems [3, 4]. The widely used message-passing mechanism is typically based on the homophily assumption [5], which assumes that nodes are inclined to behave similarly to their neighbors [6]. However, this assumption is often violated in real-world graphs, where connected nodes exhibit a contrasting tendency called *heterophily* [7]. In response to the challenges of heterophily, *heterophilic GNNs (HTGNNs)* have attracted considerable interest [6, 8, 9], with numerous innovative approaches [10–13]. However, the majority of them continue to employ a message-passing mechanism and remain successful, which was not originally designed for heterophilic graphs, as they tend to incorporate excessive messages from disparate classes. This raises a question: *Why does message passing remain effective on heterophilic graphs?*

Some efforts [6, 9, 14, 15] have begun to investigate this question and reveal that vanilla message-passing (VMP) can still work on heterophilic graphs under certain conditions. However, these works are limited in studying what kind of data is suitable for VMP. A large part of HTGNNs have modified the message-passing mechanism in response to heterophily, namely heterophilic message-passing (HTMP), which outperforms VMP on heterophilic graphs. *How HTMP works effectively in*

---

∗Corresponding Author: zhousheng_zju@zju.edu.cn.

39th Conference on Neural Information Processing Systems (NeurIPS 2025).

*certain data remains largely unexplored.* In this paper, we investigate the connection between node discriminability and the compatibility matrix (CM), that is, the latent connection preference among classes within a graph. We propose the Compatibility Matrix Discriminability (CMD) to measure the discriminability of CM. Theoretical and empirical analyses prove that node discriminability and CMD are positively correlated in VMP while ignoring the influence of node degrees. In addition, a toy example and a posterior evaluation in representative HTGNN show that *the effectiveness of existing HTMP mechanisms may be attributed to increasing CMD*, which leads to better node embeddings.

This discovery explains the essence of HTMP and provides insight for the design of new HTGNNs. Nevertheless, the graph *sparsity* and *noise* bring significant challenges for leveraging the CM as they can cause low-quality CM estimations, misguiding the message passing. To fill this gap, we propose a novel Compatibility Matrix-aware Graph Neural Network (CMGNN), which alleviates the sparsity and noise issues by introducing supplementary CM-aware messages while preserving the original neighborhood information. Meanwhile, a targeted constraint is applied to explicitly and simultaneously enhance the CMD and node embeddings. We then conduct fair comparisons to evaluate the effectiveness of CMGNN, compared with 20 baseline methods on 13 datasets with varying homophily levels and scales. Extensive experimental results demonstrate that CMGNN outperforms all baseline methods on heterophilic graphs while also being competitive on homophilic graphs. The contributions of this paper are summarized as follows:

- **Theoretical Findings**. We reveal the possible principle behind the heterophilic message-passing mechanism through comprehensive theoretical and empirical analyses, which may help people better understand the HTMP mechanism.
- **Proposed Method**. We introduce CMGNN, a novel approach that leverages and enhances the CM to learn better node embeddings while alleviating the issues of graph sparsity and noise.
- **Benchmarking and Evaluation**. We construct a comprehensive and fair benchmark to evaluate the effectiveness of CMGNN. Extensive experimental results show the superiority of CMGNN. Our code is available at https://github.com/zfx233/CMGNN.

## 2 Preliminaries

**Notations.** Given a graph $\mathcal{G} = (\mathcal{V}, \mathcal{E}, \mathbf{X}, \mathbf{A}, \mathbf{Y})$, $\mathcal{V}$ is the node set and $\mathcal{E}$ is the edge set. Nodes are characterized by the feature matrix $\mathbf{X} \in \mathbb{R}^{N \times d_f}$, where $N = |\mathcal{V}|$ denotes the number of nodes, $d_f$ is the feature dimension. $\mathbf{Y} \in \mathbb{R}^{N \times 1}$ is the node labels with the one-hot version $\mathbf{C} \in \mathbb{R}^{N \times K}$, where $K$ is the number of node classes. $\mathbf{A} \in \mathbb{R}^{N \times N}$ is the adjacency matrix, each element $a_{ij}$ in $\mathbf{A}$ denotes whether there is an edge between $i$ and $j$, and $\mathbf{D} = \text{diag}(d_1, ..., d_n)$ represents the diagonal degree matrix, where $d_i = \sum_j a_{ij}$. The normalized adjacency matrix is denoted by $\hat{\mathbf{A}} = \mathbf{D}^{-1}\mathbf{A}$. We use $\mathbf{1}$ to represent a matrix with all elements equal to 1.

**Homophily and Heterophily.** Graphs exhibit high homophily when a large fraction of neighboring nodes have the same labels as the central nodes, whereas graphs with high heterophily show the contrary. For measuring the homophily level, two widely used metrics are edge homophily $h^e$ [16] and node homophily $h^n$ [17], defined as $h^e = \frac{|\{e_{uv}|e_{uv} \in \mathcal{E}, \mathbf{Y}_u = \mathbf{Y}_v\}|}{|\mathcal{E}|}$ and $h^n = \frac{1}{|\mathcal{V}|} \sum_{v \in \mathcal{V}} \frac{|\{u|e_{uv} \in \mathcal{E}, \mathbf{Y}_u = \mathbf{Y}_v\}|}{\mathbf{d}_v}$. Both metrics range from $[0, 1]$, with higher values denoting greater homophily and lower values signifying stronger heterophily.

**Vanilla Message-Passing (VMP).** The vanilla message-passing mechanism plays a pivotal role in transforming and updating node embeddings based on the neighborhood [18]. Typically, the mechanism operates iteratively and comprises two stages:

$$\widetilde{\mathbf{Z}}^l = \text{AGGREGATE}(\mathbf{A}, \mathbf{Z}^{l-1}), \quad \mathbf{Z}^l = \text{COMBINE}\left(\mathbf{Z}^{l-1}, \widetilde{\mathbf{Z}}^l\right), \tag{1}$$

where the AGGREGATE function first aggregates the input messages $\mathbf{Z}^{l-1}$ from neighborhood $\mathbf{A}$ into $\widetilde{\mathbf{Z}}^l$, and subsequently, the COMBINE function combines the messages of the node ego and neighborhood aggregation, resulting in updated embeddings $\mathbf{Z}^l$.

**Heterophilic Message-Passing (HTMP).** To adapt to heterophily, many heterophilic message-passing methods have been proposed, which usually extend the neighborhood variously, enriching the sources of messages. In addition, the aggregation weights and the COMBINE function are

often redesigned to collect neighbor messages preferentially while preserving the ego messages. Moreover, there may exist multiple messages collected from different neighborhoods in HTMP, which simultaneously form node embeddings. In this paper, we use "HTMP" methods to refer to the heterophilic GNNs that follow the message-passing mechanism. More detailed related works are available in Appendix A.

## 3 Exploring the Essence of Message Passing on Heterophilic Graphs

In this section, we aim to uncover the underlying principle that explains the effectiveness of HTMP Inspired by the key factors identified by Zhu et al. [9], we consider the compatibility matrix (CM) [7] as a potential path. To formally explore the connection between node discriminability and CM in message passing, we use the Expected Negative KL-divergence (ENKL) [15] to measure node discriminability and define Compatibility Matrix Discriminability (CMD) for CM.

*Expected Negative KL-divergence (ENKL)* [15]. ENKL can measure the inter-class node discriminability in the graph generated by the Contextual Stochastic Block Model for Homophily/Heterophily (CSBM-H) [15], a variation of the generative model CSBM [19], which has been widely adopted to study the behavior of GNNs [6, 20, 21]. In CSBM-H, the generated graph comprises two disjoint node sets, $i \in \mathcal{C}_0$ and $j \in \mathcal{C}_1$, representing two classes. The node features are generated independently, where $x_i$ is generated from $N(\boldsymbol{\mu}_0, \sigma_0^2 \mathbf{I})$ and $x_j$ from $N(\boldsymbol{\mu}_1, \sigma_1^2 \mathbf{I})$, with $\boldsymbol{\mu}_0, \boldsymbol{\mu}_1 \in \mathbb{R}^{F_h}$ and $F_h$ as the embedding dimension. Node degrees for $\mathcal{C}_0$ and $\mathcal{C}_1$ are $d_0, d_1$ respectively. For $i \in \mathcal{C}_0$, its neighbors are independently sampled as $h \cdot d_0$ intra-class and $(1-h) \cdot d_0$ inter-class nodes, mirrored for $j \in \mathcal{C}_1$. Then, the ENKL is defined as follows:

$$\text{ENKL}(\mathbf{X}) = -d_{\mathbf{X}}^2 \left( \frac{1}{4\sigma_1^2} + \frac{1}{4\sigma_0^2} \right) - \frac{F_h}{4} \left( \rho^2 + \frac{1}{\rho^2} - 2 \right), \tag{2}$$

where $d_{\mathbf{X}}^2 = (\boldsymbol{\mu}_0 - \boldsymbol{\mu}_1)^T (\boldsymbol{\mu}_0 - \boldsymbol{\mu}_1)$, $\rho = \frac{\sigma_0}{\sigma_1}$. Different message-passing mechanisms can obtain different node embeddings with corresponding ENKL values, where smaller ENKL values correspond to more distinguishable node embeddings. Thus, ENKL can serve as a theoretically guaranteed evaluation criterion for different message passing approaches.

*Compatibility Matrix (CM)* [7]. CM characterizes the latent connection preference among classes within a graph. It is formatted as a matrix $\mathbf{M} \in \mathbb{R}^{K \times K}$, where the $i$-th row $\mathbf{M}_i$ denotes the connection probabilities between class $i$ and all classes. It can be estimated empirically as follows:

$$\mathbf{M} = \text{Norm}(\mathbf{C}^T \mathbf{C}^{nb}), \quad \mathbf{C}^{nb} = \hat{\mathbf{A}} \mathbf{C}, \tag{3}$$

where $\text{Norm}(\cdot)$ is the L1 normalization for matrix row vectors and $T$ is the matrix transpose operation. $\mathbf{C}^{nb} \in \mathbf{R}^{N \times K}$ is the *semantic neighborhoods* of nodes, indicating the proportion of neighbors from each class in the neighborhoods. Thus, the CM under CSBM-H settings is:

$$\mathbf{M}_{\text{CSBM-H}} = \begin{bmatrix} h, & 1-h \\ 1-h, & h \end{bmatrix}. \tag{4}$$

To measure the discriminability of CM, we define the Compatibility Matrix Discriminability (CMD).

**Definition 3.1** (**Compatibility Matrix Discriminability (CMD)**). Given a compatibility matrix $\mathbf{M}$, the CMD is defined as the average L1 distance between rows (classes):

$$\text{CMD}(\mathbf{M}) = \frac{\sum_i^K \sum_{j \neq i}^K \|\mathbf{M}_i - \mathbf{M}_j\|_1}{K \cdot (K-1)}, \tag{5}$$

where $K$ is the number of classes and $\mathbf{M}_i$ is the $i$-th row of $\mathbf{M}$. $\| \cdot \|_1$ denotes the L1 distance. The higher the CMD, the more distinguishable each row in the CM is.

We start with VMP to investigate the potential connection between CMD and ENKL. Following previous work [15], we use a low-pass (LP) operator $\hat{\mathbf{A}}$ as an example of VMP: $\mathbf{H}^{\text{LP}} = \hat{\mathbf{A}} \mathbf{X}$. Since all neighbors of each node share the same weight in VMP, CM can also indicate the proportion of class-level neighbor messages in node embeddings. Thus, discriminability in CM can be transferred to node embeddings in VMP. Formally, through theoretical analysis, we have the following theorem.

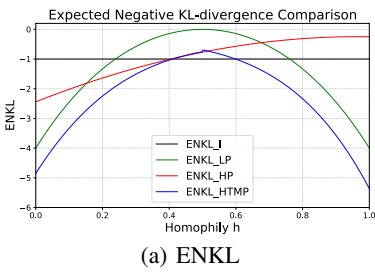
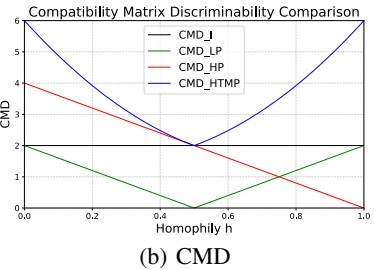

|(a) ENKL|(b) CMD|

Figure 1: Visualization of CSBM-H ($\boldsymbol{\mu}_0 = [-1, 0], \boldsymbol{\mu}_1 = [0, 1], \sigma_0^2 = 1, \sigma_1^2 = 2, d_0 = 5, d_1 = 5$).

**Theorem 3.2** (**Negatively Correlation between CMD and ENKL of VMP**). *Consider a graph* $\mathcal{G} \sim CSBM\text{-}H(\boldsymbol{\mu}_0, \boldsymbol{\mu}_1, \sigma_0^2 I, \sigma_1^2 I, d_0, d_1, h)$ *and the LP operator* $\hat{\mathbf{A}}$ *as VMP. When ignoring the influence of node degrees, i.e.,* $d_0 = d_1$, *we have* $Cov(CMD(\mathbf{M}_{CSBM\text{-}H}), ENKL(\mathbf{H}^{LP})) < 0$, *where* $Cov(\cdot)$ *denotes the covariance. It means that the CMD and ENKL of VMP are negatively correlated.*

The detailed proof is available in Appendix B. We visualize the changing trends of CMD and ENKL of VMP (LP) in Figure 1 , which does show a negative correlation. Thus, more discriminable CM can lead to better node embeddings in VMP. This gives the reason why VMP can work well in some heterophilic situations: *A CM with low homophily but high discriminability can also lead to high node discriminability in VMP.* We conduct experiments on the synthetic dataset to provide empirical analysis in Appendix C. This conclusion is similar to previous works [6, 9] that discuss heterophily from the perspective of **data**, since heterophily and CM are both characteristics of data. However, they lack the understanding from the perspective of the model, which is conducive to guiding the designs of HTGNNs. To fill this gap, we take a further step in understanding this theoretical finding with a **model** perspective and consider such a question: *when given the exact data, can we enhance the CMD through the model to learn better embeddings?* Coincidentally, this may able to explain the essence of HTMP methods, as the weights of practical class-level neighbor messages in these HTMP methods may be different from the original CM. Since CM is a fixed concept for describing the original data, we introduce Weighted-CM for the sake of distinction.

**Definition 3.3** (**Weighted Compatibility Matrix (Weighted-CM)**). The weights of practical class-level neighbor messages after message passing. It is also formatted as a matrix $\tilde{\mathbf{M}} \in \mathbb{R}^{K \times K}$, where the $i$-th row $\tilde{\mathbf{M}}_i$ denotes the aggregate weights of all classes for class $i$. It can be estimated as follows:

$$\tilde{\mathbf{M}} = \mathbf{C}^T \dot{\mathbf{A}} \mathbf{C}, \tag{6}$$

where $\dot{\mathbf{A}} \in \mathbb{R}^{N \times N}$ is the practical aggregate weights matrix during message passing. Note that there is no strict limit to the values: $\tilde{m}_{ij} \in \mathbb{R}$.

For further analysis, we define the identity (I), high-pass (HP) and heterophilic message-pass (HTMP) operators as $\dot{\mathbf{A}}^{\mathrm{I}} = \mathbf{I}$, $\dot{\mathbf{A}}^{\mathrm{HP}} = \mathbf{I} - \hat{\mathbf{A}}$, and $\dot{\mathbf{A}}^{\mathrm{HTMP}} = a\mathbf{I} + b\hat{\mathbf{A}} + c\hat{\mathbf{A}}^2$, where $a, b, c \in [-1, 1]$ are three learnable weight parameters. The high-pass operator can capture the difference between the central node and its neighborhoods in heterophilic graphs. The HTMP operator is a typical scheme that aims to utilize the linear combination of multiple-order neighborhood messages to obtain more distinguishable embeddings. The embedding matrices after message passing can be represented as $\mathbf{H}^{\mathrm{o}} = \dot{\mathbf{A}}^{\mathrm{o}} \mathbf{X}, \mathrm{o} \in \{\mathrm{I, LP, HP, HTMP}\}$. Thus, the Weighted-CMs and corresponding CMDs of identity, LP, HP, and HTMP operators are as follows:

$$\tilde{\mathbf{M}}^{\mathrm{I}} = \begin{bmatrix} 1, & 0 \\ 0, & 1 \end{bmatrix}, \ \tilde{\mathbf{M}}^{\mathrm{LP}} = \begin{bmatrix} h, & 1-h \\ 1-h, & h \end{bmatrix}, \ \tilde{\mathbf{M}}^{\mathrm{HP}} = \begin{bmatrix} 1-h, & h-1 \\ h-1, & 1-h \end{bmatrix},$$

$$\tilde{\mathbf{M}}^{\mathrm{HTMP}} = \begin{bmatrix} a + bh + c[h^2 + (1-h)^2], & b(1-h) + 2ch(1-h) \\ b(1-h) + 2ch(1-h), & a + bh + c[h^2 + (1-h)^2] \end{bmatrix}, \tag{7}$$

$\mathrm{CMD}^{\mathrm{I}} = 2, \ \mathrm{CMD}^{\mathrm{LP}} = 4|h - \frac{1}{2}|, \ \mathrm{CMD}^{\mathrm{HP}} = 4|h - 1|, \ \mathrm{CMD}^{\mathrm{HTMP}} = 4|\frac{a}{2} + (h - \frac{1}{2})b + 2(h - \frac{1}{2})^2 c|.$

Table 1: CMD and ENKL values of different operators on real-world graphs.

| Dataset | Roman-Empire | | Amazon-Ratings | | Chameleon-F | | Squirrel-F | |
|---|---|---|---|---|---|---|---|---|
| Operator | CMD | ENKL | CMD | ENKL | CMD | ENKL | CMD | ENKL |
| LP | 0.85 | -0.76 | 0.46 | -0.26 | 0.27 | -25.38 | 0.08 | -14.84 |
| HP | 2.77 | -2.62 | 1.95 | -0.80 | 2.07 | -126.5 | 2.02 | -67.89 |
| HTMP (Case 1) | 3.31 | -4.13 | 2.35 | -1.14 | 2.39 | -150.8 | 2.22 | -80.05 |
| HTMP (Case 2) | 3.55 | -5.04 | 3.10 | -2.41 | 2.85 | -170.6 | 2.28 | -89.52 |

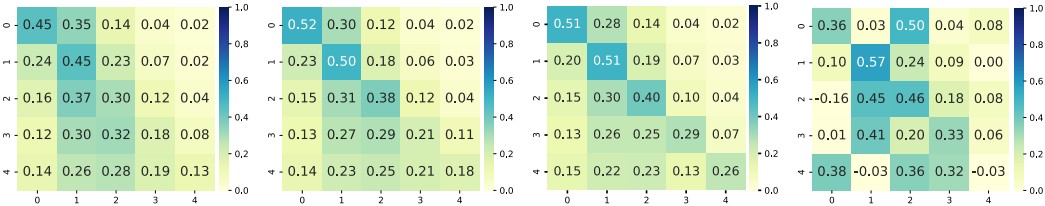

(a) Fixed(LP),CMD=0.46    (b) GloGNN,CMD=0.59    (c) GPR-GNN,CMD=0.62    (d) ACM-GCN,CMD=0.97

Figure 2: Visualizations of Weighted-CMs estimated through various methods on Amazon-Ratings.

Since we aim to obtain the largest CMD, the values of $a, b, c$ can be set as follows:

$$\text{Case 1, } h \geq \frac{1}{2}: a = 1, b = 1, c = 1, \dot{\mathbf{A}}^{\text{HTMP}} = \mathbf{I} + \hat{\mathbf{A}} + \hat{\mathbf{A}}^2, \text{CMD}^{\text{HTMP}} = 8(h - \frac{1}{4})^2 + \frac{3}{2},$$

$$\text{Case 2, } h < \frac{1}{2}: a = 1, b = -1, c = 1, \dot{\mathbf{A}}^{\text{HTMP}} = \mathbf{I} - \hat{\mathbf{A}} + \hat{\mathbf{A}}^2, \text{CMD}^{\text{HTMP}} = 8(h - \frac{3}{4})^2 + \frac{3}{2}.$$

$$(8)$$

We visualize the changing trends of CMD and ENKL of the four operators in Figure 1. Similarly, the two changing trends are negatively correlated independently for each operator, which can also be proven by theoretical analysis. Although the relationship between the CMD and ENKL values among different operators is not consistent, we still show that *it is feasible to obtain better embeddings by increasing the CMD in HTMP as the HTMP operator achieves the minimum ENKL*, i.e., the best node discriminability, in most cases by different combinations of parameters $a, b, c$.

However, there are significant gaps between the theoretical assumptions of CSBM-H and real-world graphs, including multi-class, semi-supervised settings, imbalanced node degrees, and potential low-quality node features, among others. *What is the impact on the relationship between CMD and ENKL?* We count the values of ENKL and CMD when different operators are used on some real graphs, as in Table 1. We find that *the negative correlation between CMD and ENKL still holds on real-world graphs*, which expands the Theorem 3.2. In addition to the above analyses, we also investigated three representative methods, GloGNN [22], GPR-GNN [23], and ACM-GCN [12].

**Observation 3.4** (**The increased CMDs of representative HTGNNs**). We conduct a posterior evaluation on the results of GloGNN, GPR-GNN, and ACM-GCN. All the CMDs of Weighted-CMs have been significantly improved compared to the fixed one of the original data, as shown in Figure 2.

More details and examples about the posterior evaluation can be found in Appendix D. The success of the HTMP operator and Observation 3.4 gives a possible answer to the effectiveness of HTMP:

**Remark 3.5** (**Increasing CMD is a possible reason for the effectiveness of HTMP**). Given certain data, the HTMP mechanism can increase the CMD by replacing the fixed one with the Weighted-CM via various model-level designs, leading to better node embeddings.

This explains the underlying principle of HTMP and provides insight for designing new HTGNNs via increasing CMD. Several existing works [7, 24, 25] have naively estimated and used CM without increasing CMD. However, to make good use of CM, there are still two significant issues, namely sparsity and noise, when considering real-world heterophilic graphs. First, *node degrees can be very imbalanced*. Some nodes may have few neighbors due to the sparsity of the graph, which may not only show a highly inconsistent semantic neighborhood with CM but also be harmful to the effectiveness of message passing. Second, *node labels are not fully available* in the semi-supervised

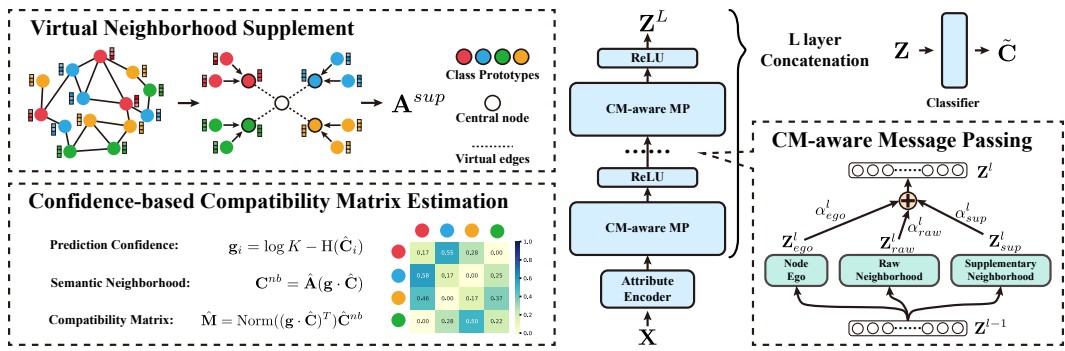

Figure 3: The overall framework of CMGNN.

setting. Inaccurate predictions of node labels can cause low-quality CM estimation, which misguides the message passing, leading to low-discriminative node embeddings. These issues pose significant challenges to the explicit modeling and effective utilization of CM.

## 4  Methodology

To address these issues, we introduce Compatibility Matrix-Aware GNN (CMGNN), as shown in Figure 3, which alleviates the sparsity and noise issues by *virtual neighborhood supplement*, *confidence-based compatibility matrix estimation*, and *CM-aware message passing*. Then, a straightforward constraint is applied to explicitly enhance the discriminability of CM and node embeddings.

**Virtual Neighborhood Supplement.** Given the graph sparsity, some nodes may have low degrees. Thus, CMGNN introduces an additional virtual neighborhood to provide nodes with supplemental messages from each class. The virtual neighborhood is the same for all nodes, which consists of $K$ additional virtual neighbors: $\mathbf{A}^{sup} = \mathbf{1} \in \mathbb{R}^{N \times K}$. It ensures accessibility to the messages of all classes for each node. Specifically, these additional neighbors are $K$ virtual nodes, designed as the class prototypes based on the training set labels. The attributes $\mathbf{X}^{ptt} \in \mathbb{R}^{K \times d_f}$, neighborhoods $\mathbf{A}^{ptt} \in \mathbb{R}^{K \times N}$ and labels $\mathbf{Y}^{ptt} \in \mathbb{R}^{K \times K}$ of prototypes are defined as follows:

$$\mathbf{X}^{ptt} = \text{Norm}(\mathbf{C}_{train}{}^T \mathbf{X}_{train}), \ \mathbf{A}^{ptt} = \mathbf{0}, \ \mathbf{Y}^{ptt} = \mathbf{I}, \tag{9}$$

where $\mathbf{C}_{train}$ and $\mathbf{X}_{train}$ are the one-hot labels and attributes of nodes in the training set. Utilizing class prototypes to construct virtual neighborhoods can provide each node with representative messages of all classes, which builds the basis for CM-aware message passing.

**Confidence-based CM Estimation.** The CM can be calculated via Eq 3 with full-available labels. However, label information is not entirely available in semi-supervised settings. Thus, we try to estimate the CM with the help of pseudo labels. To mitigate the impact of potentially incorrect pseudo labels predicted by the model, we introduce a confidence metric $\mathbf{g} \in \mathbb{R}^{N \times 1}$ derived from the information entropy, with high entropy indicating low confidence:

$$\mathbf{g}_i = \log K - \text{H}(\hat{\mathbf{C}}_i) \in [0, \log K], \tag{10}$$

where $\text{H}(p) = -\sum_i p_i \log(p_i)$ denotes the entropy, $\hat{\mathbf{C}} \in \mathbb{R}^{N \times K}$ is the soft pseudo labels composed of training labels $\mathbf{C}_{train}$ and model predictions $\tilde{\mathbf{C}}$:

$$\hat{\mathbf{C}}_i = \begin{cases} \mathbf{C}_{train,i}, & v_i \in \mathcal{V}_{train}, \\ \tilde{\mathbf{C}}_i, & \text{otherwise}, \end{cases} \tag{11}$$

where $\mathcal{V}_{train}$ is the training set. Then the semantic neighborhoods of the nodes are calculated considering the confidence: $\hat{\mathbf{C}}^{nb} = \text{Norm}(\hat{\mathbf{A}}(\mathbf{g} \cdot \hat{\mathbf{C}})) \in \mathbb{R}^{N \times K}$. Finally, we can estimate the compatibility matrix $\hat{\mathbf{M}} \in \mathbb{R}^{K \times K}$ as follows:

$$\hat{\mathbf{M}} = \text{Norm}((\mathbf{g} \cdot \hat{\mathbf{C}})^T)\hat{\mathbf{C}}^{nb}. \tag{12}$$

Note that the CM is repeatedly updated during training. To improve efficiency and stability, CM is not estimated in every epoch. It remains fixed until the evaluation performance improves.

**CM-aware Message Passing.** Relying entirely on CM to guide message passing can lead to confusing embeddings when the quality of CM is low. Therefore, we regard it as a separate supplementary message source while preserving the original neighborhood distribution. Specifically, CMGNN collects messages from node ego, raw neighborhoods $\mathbf{A}$ and the supplement neighborhoods $\mathbf{A}^{sup}$, respectively. The first two are widely used in HTGNNs as the ego-neighbor separation design principle [16], while the last provides a novel approach to apply CM. The messages from node ego and raw neighborhoods are obtained as follows:

$$\mathbf{Z}_{ego}^{l} = \mathbf{Z}^{l-1}\mathbf{W}_{ego}^{l}, \ \ \mathbf{Z}_{raw}^{l} = \hat{\mathbf{A}}\mathbf{Z}^{l-1}\mathbf{W}_{raw}^{l}, \tag{13}$$

where $\mathbf{Z}^{l-1}$ is the input node embedding of layer $l$, $\mathbf{W}_{ego}^{l}, \mathbf{W}_{raw}^{l}$ are learnable matrices for message transformation. For the supplement neighborhood, we leverage CM to offer nodes additional messages personalized by their soft pseudo labels, which converts discriminability from CM into messages:

$$\mathbf{Z}_{sup}^{l} = (\mathbf{A}^{sup} \odot \hat{\mathbf{C}}\hat{\mathbf{M}})\mathbf{Z}_{ptt}^{l-1}\mathbf{W}_{sup}^{l}, \tag{14}$$

where $\odot$ is the Hadamard product, $\mathbf{Z}_{ptt}^{l-1}$ are the input embeddings of virtual prototype nodes obtained via the same message-passing process as real nodes. The aggregation weights $\hat{\mathbf{C}}\hat{\mathbf{M}}$ indicate the desired semantic neighborhoods of nodes, i.e., the desired class proportion of neighbors according to the class probability of central nodes. Using soft logits rather than one-hot pseudo labels preserves the real characteristics of nodes and reduces the impact of wrong predictions.

Given the diverse conditions of nodes, we apply adaptive weighted addition to combine these messages. Meanwhile, messages of multiple layers are concatenated to preserve information from different localities within the graph. The overall CM-aware message passing is described as follows:

$$\mathbf{Z}^{l} = \mathrm{diag}(\alpha_{ego}^{l})\mathbf{Z}_{ego}^{l} + \mathrm{diag}(\alpha_{raw}^{l})\mathbf{Z}_{raw}^{l} + \mathrm{diag}(\alpha_{sup}^{l})\mathbf{Z}_{sup}^{l}, \quad \mathbf{Z} = \overset{L}{\underset{l=0}{\|}} \mathbf{Z}^{l}, \tag{15}$$

where $\mathrm{diag}(\alpha_{ego}^{l}), \mathrm{diag}(\alpha_{raw}^{l}), \mathrm{diag}(\alpha_{sup}^{l}) \in \mathbb{R}^{N \times 1}$ are the learnable combination weights, $\|$ denotes the concatenation operation, $\mathbf{Z}$ is the final node embeddings. The input embedding of the first layer is $\mathbf{Z}^{0} = \mathbf{X}\mathbf{W}^{0}$ where $\mathbf{W}^{0} \in \mathbb{R}^{d_f \times d_r}$ and $d_r$ is the dimension of node emebddings. In practice, we use ReLU as the activation function between layers. Note that we use the prediction of the model $\tilde{\mathbf{C}}$ to estimate CM in the above process. It is initialized as a uniform distribution on each class and replaced by the output of CMGNN via a classifier CLA during the learning process:

$$\tilde{\mathbf{C}} = \mathrm{CLA}(\mathbf{Z}). \tag{16}$$

**Objective Function.** As mentioned in Sec 3, enhancing the discriminability of CM is beneficial for learning better node embeddings. Thus, we introduce an additional discrimination loss $\mathcal{L}_{dis}$ to reduce the similarity of the desired neighborhood message among different classes, which simultaneously enhances the discriminability of CM and node embeddings. Experimental analysis in Section 5.4 demonstrates the effectiveness of this constraint to enhance the CM. The overall loss consists of a CrossEntropy loss $\mathcal{L}_{ce}$ and the discrimination loss $\mathcal{L}_{dis}$:

$$\mathcal{L} = \mathcal{L}_{ce}(\tilde{\mathbf{Z}}, \mathbf{Y}) + \lambda\mathcal{L}_{dis}, \quad \mathcal{L}_{dis} = \sum_{i \neq j} \mathrm{Sim}(\hat{\mathbf{M}}_i\mathbf{Z}_{ptt}, \hat{\mathbf{M}}_j\mathbf{Z}_{ptt}), \tag{17}$$

where $\mathrm{Sim}(\cdot)$ denotes the cosine similarity, $\mathbf{Z}_{ptt} \in \mathbb{R}^{K \times d_r}$ is the embeddings of virtual prototype nodes. More details of CMGNN including pseudo-code are available in Appendix E.

# 5 Experiments

## 5.1 Experimental Settings

As reported in Platonov et al. [26], some widely adopted datasets in existing works have critical drawbacks, leading to unreliable comparisons. Therefore, with a comprehensive review of existing benchmark evaluations, we construct a new collection of datasets and a unified codebase to fairly perform experimental evaluation. Specifically, we integrate 20 representative homophilic and heterophilic GNNs, construct a unified codebase, and evaluate their node classification performances on 13 datasets with various scales and heterophily levels.

Table 2: Node classification accuracy comparison (%). The error bar (±) denotes the standard deviation of results over 10 runs. The best and second-best results in each column are highlighted in **bold** font and underlined. OOM denotes out-of-memory error during training.

| Dataset | Roman-Empire | Amazon-Ratings | Chameleon-F | Squirrel-F | Actor | Flickr | BlogCatalog | Pubmed | Avg. Rank |
|---|---|---|---|---|---|---|---|---|---|
| **Homo.** | 0.05 | 0.38 | 0.25 | 0.22 | 0.22 | 0.24 | 0.4 | 0.8 | |
| **Nodes** | 22,662 | 24,492 | 890 | 2,223 | 7,600 | 7,575 | 5,196 | 19,717 | |
| **Edges** | 65,854 | 186,100 | 13,584 | 65,718 | 30,019 | 479,476 | 343,486 | 88,651 | |
| **Classes** | 18 | 5 | 5 | 5 | 5 | 9 | 6 | 3 | |
| **MLP** | 62.29 ± 1.03 | 42.66 ± 0.84 | 38.66 ± 4.02 | 36.74 ± 1.80 | 36.74 ± 0.85 | 89.82 ± 0.63 | 93.57 ± 0.55 | 87.48 ± 0.46 | 15.6 |
| GCN | 38.58 ± 2.35 | 45.00 ± 0.55 | 42.12 ± 3.82 | 37.89 ± 2.40 | 30.09 ± 0.74 | 68.33 ± 2.82 | 78.07 ± 1.17 | 87.66 ± 0.42 | 16.3 |
| GAT | 59.55 ± 1.45 | 47.72 ± 0.73 | 40.89 ± 3.50 | 38.22 ± 1.71 | 30.94 ± 0.95 | 57.22 ± 3.04 | 88.36 ± 1.37 | 87.45 ± 0.53 | 15.6 |
| GraphSAGE | 69.62 ± 1.40 | 45.07 ± 0.54 | 42.18 ± 4.64 | 38.13 ± 1.71 | 36.12 ± 1.40 | 92.00 ± 0.58 | 96.30 ± 0.44 | 88.86 ± 0.56 | 9.0 |
| APPNP | 70.77 ± 0.66 | 45.97 ± 0.49 | 42.07 ± 4.07 | 36.38 ± 1.20 | 34.86 ± 1.32 | 91.50 ± 0.51 | 96.29 ± 0.41 | 89.22 ± 0.58 | 10.9 |
| GCNII | 82.53 ± 0.37 | 47.53 ± 0.72 | 41.56 ± 4.15 | 40.70 ± 1.80 | **37.51 ± 0.92** | 91.64 ± 0.67 | 96.48 ± 0.62 | 89.96 ± 0.43 | 4.9 |
| H2GCN | 68.61 ± 1.05 | 37.20 ± 0.67 | 42.29 ± 4.57 | 35.82 ± 2.20 | 33.32 ± 0.90 | 91.25 ± 0.58 | 96.24 ± 0.39 | 89.32 ± 0.37 | 12.6 |
| MixHop | 79.06 ± 0.64 | 47.41 ± 1.00 | 44.97 ± 3.12 | 40.43 ± 1.40 | 36.99 ± 0.88 | 91.10 ± 0.46 | 96.22 ± 0.42 | 89.47 ± 0.35 | 6.6 |
| GBK-GNN | 66.04 ± 1.44 | 40.18 ± 1.94 | 41.73 ± 4.57 | 36.49 ± 1.37 | 35.91 ± 0.84 | OOM | OOM | 88.14 ± 0.43 | 16.5 |
| GGCN | OOM | OOM | 41.23 ± 4.08 | 36.76 ± 2.19 | 35.68 ± 0.87 | 90.84 ± 0.65 | 95.58 ± 0.44 | 89.04 ± 0.40 | 15.4 |
| GloGNN | 68.63 ± 0.63 | 48.62 ± 0.59 | 40.95 ± 5.95 | 36.85 ± 1.97 | 36.66 ± 0.81 | 90.47 ± 0.77 | 94.51 ± 0.49 | 89.60 ± 0.34 | 11.4 |
| HOGGCN | OOM | OOM | 43.35 ± 3.66 | 38.63 ± 1.95 | 36.47 ± 0.83 | 90.94 ± 0.72 | 94.75 ± 0.65 | OOM | 13.1 |
| GPR-GNN | 71.10 ± 0.66 | 46.87 ± 0.60 | 42.85 ± 3.48 | 37.66 ± 1.08 | 36.16 ± 1.02 | 91.20 ± 0.46 | 96.29 ± 0.44 | 89.26 ± 0.37 | 8.5 |
| ACM-GCN | 71.15 ± 0.73 | 50.64 ± 0.61 | 45.20 ± 4.14 | 40.90 ± 1.74 | 35.99 ± 1.44 | 91.43 ± 0.65 | 96.16 ± 0.57 | 89.94 ± 0.35 | 6.0 |
| OrderedGNN | 82.88 ± 0.71 | 51.15 ± 0.46 | 41.51 ± 4.15 | 36.94 ± 2.94 | 37.07 ± 0.95 | 91.43 ± 0.78 | 96.22 ± 0.35 | **90.01 ± 0.35** | 6.4 |
| M2MGNN | 83.97 ± 0.71 | 50.93 ± 0.45 | 40.39 ± 3.68 | 36.36 ± 3.06 | 35.92 ± 0.76 | 91.49 ± 0.76 | 96.34 ± 0.48 | 89.91 ± 0.35 | 8.8 |
| N² | 80.42 ± 1.30 | 49.94 ± 0.86 | 42.46 ± 4.37 | 40.92 ± 2.25 | 35.51 ± 1.20 | 90.85 ± 0.78 | 96.22 ± 0.63 | 88.53 ± 0.50 | 8.5 |
| CLP | 67.36 ± 0.62 | 47.42 ± 0.44 | 41.73 ± 4.49 | 37.64 ± 1.37 | 36.67 ± 1.64 | 90.13 ± 0.67 | 94.45 ± 0.59 | 88.88 ± 0.42 | 11.8 |
| EPFGNN | 43.05 ± 0.40 | 45.16 ± 0.73 | 44.30 ± 3.91 | 40.47 ± 1.54 | 30.24 ± 1.14 | 57.54 ± 1.77 | | 87.06 ± 0.53 | 15.0 |
| CPGNN | 59.55 ± 0.84 | 46.65 ± 0.71 | 41.45 ± 4.84 | 37.24 ± 2.09 | 33.37 ± 1.02 | 80.46 ± 1.25 | 81.92 ± 1.06 | 87.98 ± 0.40 | 15.6 |
| **CMGNN** | **84.35 ± 1.27** | **52.13 ± 0.55** | **45.70 ± 4.92** | **41.89 ± 2.34** | 36.82 ± 0.78 | **92.66 ± 0.46** | **97.00 ± 0.52** | 89.99 ± 0.32 | **1.5** |

Table 3: Node classification accuracy (%) and time cost (minutes) comparison on large-scale graphs.

| Dataset | Penn94 | | Twitch-Gamer | | Genius | | Pokec | | Snap-Patents | | Avg. Rank |
|---|---|---|---|---|---|---|---|---|---|---|---|
| **Homo.** | 0.47 | | 0.55 | | 0.62 | | 0.45 | | 0.07 | | |
| **Nodes** | 41,554 | | 168,114 | | 421,961 | | 1,632,803 | | 2,923,922 | | |
| **Edges** | 1,362,229 | | 6,797,557 | | 984,979 | | 30,622,564 | | 13,975,788 | | |
| **Classes** | 2 | | 2 | | 2 | | 2 | | 5 | | |
| Method | Accuracy | Cost | Accuracy | Cost | Accuracy | Cost | Accuracy | Cost | Accuracy | Cost | |
| MLP | 74.71 ± 0.40 | 1 | 61.25 ± 0.19 | 1.5 | 82.54 ± 0.14 | 0.8 | 62.27 ± 0.08 | 48 | 31.50 ± 0.06 | 14 | 13.8 |
| GCN | 78.48 ± 0.41 | 2.7 | 61.30 ± 0.17 | 2.6 | 84.05 ± 0.12 | 1.6 | 70.17 ± 0.10 | 75 | 37.91 ± 0.06 | 35 | 9.4 |
| GAT | 77.94 ± 1.22 | 25 | 64.59 ± 0.20 | 103 | 82.01 ± 0.45 | 5 | 75.36 ± 0.18 | 243 | 38.38 ± 0.15 | 142 | 8.8 |
| GraphSAGE | 78.90 ± 0.37 | 2.8 | 62.14 ± 0.09 | 14 | 84.37 ± 0.15 | 4.1 | 77.22 ± 0.06 | 134 | 35.72 ± 0.12 | 125 | 7.8 |
| APPNP | 77.04 ± 0.35 | 4.5 | 60.28 ± 0.23 | 5.4 | 82.81 ± 0.29 | 4.7 | 62.01 ± 3.83 | 34 | 32.64 ± 0.07 | 57 | 13.4 |
| GCNII | 79.97 ± 0.40 | 14 | 64.97 ± 0.40 | 221 | **86.45 ± 0.19** | 89 | 77.75 ± 1.17 | 343 | 42.24 ± 0.68 | 495 | 3.4 |
| MixHop | 76.74 ± 1.17 | 8 | 62.25 ± 0.17 | 52 | 83.49 ± 0.27 | 10 | 76.97 ± 0.27 | 305 | 36.94 ± 0.17 | 152 | 8.6 |
| GloGNN | 82.29 ± 0.62 | 8 | 65.74 ± 0.20 | 163 | 84.54 ± 0.15 | 9.5 | 78.61 ± 0.79 | 263 | 34.27 ± 5.28 | 165 | 4.0 |
| GPR-GNN | 82.50 ± 0.32 | 9 | 61.03 ± 0.38 | 8 | 82.89 ± 0.64 | 4.4 | 75.45 ± 0.08 | 54 | 32.87 ± 0.10 | 39 | 9.2 |
| ACM-GCN | 80.67 ± 0.49 | 12 | 61.87 ± 0.71 | 58 | 80.63 ± 0.53 | 4.9 | 74.60 ± 0.46 | 393 | 37.53 ± 0.22 | 378 | 10.2 |
| OrderedGNN | 79.49 ± 0.86 | 14 | 64.55 ± 0.32 | 66 | 84.83 ± 0.75 | 31 | 75.79 ± 0.20 | 265 | 39.56 ± 0.34 | 385 | 5.4 |
| M2MGNN | 81.86 ± 0.24 | 74 | OOM | / | 84.43 ± 1.40 | 100 | OOM | / | OOM | / | 11.4 |
| N² | 80.69 ± 0.44 | 35 | 65.76 ± 0.19 | 294 | 84.08 ± 1.40 | 538 | OOM | / | OOM | / | 8.8 |
| CLP | 74.62 ± 0.53 | 0.5 | 63.77 ± 0.18 | 1.7 | 82.51 ± 0.14 | 1.3 | 67.23 ± 0.10 | 13 | 32.05 ± 0.06 | 20 | 12.4 |
| EPFGNN | 72.53 ± 0.58 | 12 | 64.19 ± 0.23 | 21 | 81.99 ± 0.44 | 6.5 | 64.19 ± 0.23 | / | OOM | / | 13.4 |
| CPGNN | 78.20 ± 0.42 | 14 | 63.06 ± 0.25 | 15 | 80.70 ± 0.80 | 3.9 | 75.80 ± 0.11 | 123 | 37.09 ± 0.09 | 137 | 9.8 |
| **CMGNN** | **83.01 ± 0.48** | 17 | 65.18 ± 0.31 | 247 | 85.19 ± 0.53 | 20 | **81.42 ± 0.55** | 376 | **59.86 ± 0.61** | 69 | **1.6** |

**Newly Organized Datasets.** The newly organized datasets include (i) small-scale: Roman-Empire, Amazon-Ratings, Chameleon-F, Squirrel-F, Actor, Flickr, BlogCatalog and Pubmed; (ii) large-scale: Penn94, Twitch-Gamer, Genius, Pokec and Snap-Patents. Their statistics are summarized in Table 2 and Table 3. For consistency with existing methods, we randomly construct 10 splits with predefined proportions (48% / 32% / 20% for training / validation / test) for each dataset and report the mean accuracy and standard deviation of 10 splits.

**Baseline Methods.** For baseline methods, we choose 20 representative homophilic and heterophilic GNNs, including (i) shallow base model: MLP; (ii) homophilic GNNs: GCN [1], GAT [27], Graph-SAGE [28], APPNP [29], GCNII [30]; (iii) heterophilic GNNs: H2GCN [16], MixHop [10], GBK-GNN [31], GGCN [32], GloGNN [22], HOGGCN [33], GPR-GNN [23], ACM-GCN [12], OrderedGNN [13], M2MGNN [34] and N² [35]; (iv) compatibility matrix-based methods: CLP [25], EPFGNN [24], CPGNN [7]. For each method, we integrate its code into a unified codebase and search for parameters in the space suggested by the original papers. All methods share the same call interfaces, ensuring a fair environment for comparison. More detailed descriptions about the drawbacks of previous datasets, newly organized datasets and other experimental settings can be found in Appendix F.

## 5.2 Performance Comparison

We evaluate the above methods and report their performances in Table 2 and Table 3.

Table 4: Ablation study results (%) between CMGNN and five ablation variants, where CE denotes the confidence-based CM estimation, SM denotes supplementary messages from the sufficient neighborhoods and DL denotes the discrimination loss.

| Variants | Roman-Empire | Squirrel-F | Actor | Chameleon-F | Amazon-Ratings | Pokec | Penn94 | Genius |
|---|---|---|---|---|---|---|---|---|
| CMGNN | **84.35 ± 1.27** | **41.89 ± 2.34** | **36.82 ± 0.78** | **45.70 ± 4.92** | **52.13 ± 0.55** | **81.42 ± 0.55** | **83.01 ± 0.48** | **85.19 ± 0.53** |
| W/O CE | 83.88 ± 1.41 | 40.35 ± 2.43 | 36.47 ± 1.22 | 44.75 ± 3.05 | 51.93 ± 0.38 | 80.67 ± 0.65 | 82.58 ± 0.49 | 85.03 ± 0.47 |
| W/O SM | 83.82 ± 1.29 | 40.72 ± 2.28 | 36.05 ± 1.24 | 42.29 ± 4.38 | 51.91 ± 0.83 | 79.39 ± 0.37 | 81.68 ± 1.55 | 84.94 ± 0.56 |
| W/O DL | 83.63 ± 1.35 | 41.65 ± 2.55 | 36.41 ± 1.08 | 44.92 ± 4.12 | 52.05 ± 0.57 | 80.28 ± 0.49 | 81.34 ± 1.83 | 84.87 ± 0.75 |
| W/O CE and DL | 83.77 ± 1.38 | 39.80 ± 2.36 | 36.32 ± 1.05 | 44.58 ± 3.28 | 51.74 ± 0.55 | 79.26 ± 0.34 | 81.20 ± 0.61 | 84.53 ± 0.59 |
| W/O SM and DL | 83.48 ± 1.89 | 40.19 ± 2.69 | 35.66 ± 1.42 | 41.01 ± 3.09 | 51.49 ± 1.02 | 77.57 ± 0.44 | 80.71 ± 0.48 | 84.28 ± 0.50 |

**Performance of Baselines.** With the comprehensive benchmark, some interesting observations can be found. First, comparing MLP and *homophilic GNNs* , we find that VMP can still work well in Amazon-Ratings, Chameleon-F, and Squirrel-F, which meet the observations of previous works. Specifically, GCNII achieves an average rank of 4.9, which is even better than all HTGNNs. This may be attributed to the initial residual connection in GCNII, which enhances the practical Weighted-CM in message passing. As for *heterophilic GNNs*, they generally achieve better results than VMPs (GCN, GAT), demonstrating the effectiveness of their various designs for heterophily. Notably, MixHop, as an early method, can also achieve quite good performance. The previous SOTA methods, OrderedGNN and ACM-GCN, prove their effectiveness again through good rankings.

**Performance of CMGNN.** CMGNN achieves the best performance in 6 small-scale datasets with an average rank of 1.5, outperforming all baseline methods. This demonstrates the superiority of leveraging CM and increasing CMD while alleviating the sparsity and noise issues. Regarding the suboptimal performance in Actor, we speculate that this is because its node attributes and CM are not discriminative enough to provide valuable information via the supplementary messages, and are hard to enhance. On Pubmed, the raw identity-like CM is good enough, leading to a minor contribution from supplement messages. Despite this, CMGNN still achieves top-level performance.

**Comparision with CM-based Methods.** Some existing methods also utilize the CM to redefine pairwise relations (e.g., edge weights) for nodes. However, they suffer from the issues of sparsity and noise, as their performances and average ranks have significant gaps compared with CMGNN. In contrast, CMGNN leverages CM and virtual neighbors to construct supplementary messages while preserving the original neighborhood distribution, leading to the following advantages: (i) Better robustness for low-quality pseudo labels; (ii) Unlock the effectiveness of CM for low-degree nodes; (iii) More accurate estimation of CM. More detailed analyses are listed in Appendix G.1.

**Performance on Large-Scale Graphs.** To further evaluate the scalability of CMGNN, we also conduct experiments on five large-scale datasets. The performance and computational cost comparison are listed in Table 3. Again, CMGNN achieves superior performance with the best average rank of 1.6, while GCNII follows behind with a rank of 3.4. Meanwhile, CMGNN strikes a good balance between performance and efficiency, especially in Snap-Patents with 17% significant improvements and less time cost, demonstrating great scalability in handling of large-scale graphs.

**Performance on Homophilic and Heterophilic Graphs.** We divide all datasets into two groups according to their edge homophily levels with a threshold of 0.5. CMGNN shows significant effectiveness on heterophilic graphs with an average rank of 1.3 and achieves the best on 9 of 10 datasets. Also, CMGNN can keep competitive performance on homophilic graphs with an average rank of 2.3, which is also the best compared with baseline methods. Interestingly, some heterophilic GNNs work relatively better on homophilic graphs rather than heterophilic graphs, such as GloGNN and OrderedGNN. This might be because these methods are relatively more inclined to adapt to both situations.

## 5.3 Ablation Study

We conduct an ablation study on three key designs of CMGNN, including the confidence-based CM estimation (CE), supplementary messages from the virtual neighborhood (SM) and the discrimination loss (DL). The results are shown in Table 4. *Firstly*, all three components have indispensable contributions to CMGNN, as the absence of any part will degrade the performance. *Meanwhile*, the CE and DL have relatively small impacts while SM plays a more important role in most datasets. *Further*, we notice that CMGNN can reach a smaller standard deviation compared with the variants

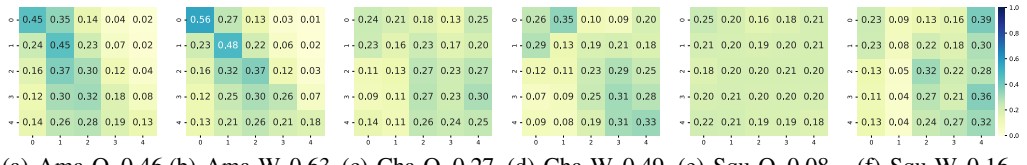

(a) Ama-O, 0.46 (b) Ama-W, 0.63 (c) Cha-O, 0.27 (d) Cha-W, 0.49 (e) Squ-O, 0.08 (f) Squ-W, 0.16

Figure 4: Visualizations of the original CM (O) and Weighted-CMs (W) on Amazon-Ratings (Ama), Chameleon-F (Cha), and Squirrel-F (Squ), along with the corresponding CMD values.

most of the time. This shows that CMGNN achieves more stable results by alleviating the sparsity and noise issues. As for the opposite result on Chameleon-F, this may be attributed to its small size (890 nodes), which naturally leads to unstable results.

### 5.4 Case Study on the Estimated CMs and CMDs

To evaluate the effectiveness of CMGNN in increasing CMD, we calculate the CMD of both original and Weighted-CMs, as illustrated in Figure 4. The results demonstrate that CMGNN is capable of enhancing the CMD across graphs with varying homophily levels, thereby resulting in better node embeddings. Even for CM with tiny CMD on Squirrel-F, CMGNN can still maintain its effectiveness.

More experimental results can be found in Appendix G, including more ablation studies, case studies on node degrees and low label rate settings, comprehensive complexity analysis, and runtime-performance tradeoff comparison.

## 6 Conclusion and Limitation

In this paper, we explore the underlying principle that explains the effectiveness of HTMP by investigating the connection between node discriminability and CM. We find that the effectiveness of many existing HTMP mechanisms may be credited to increasing the CMD. Inspired by this discovery, we propose CMGNN, a novel approach to enhance the CMD and node embeddings explicitly while alleviating the sparsity and noise issues. Experimental results show the effectiveness of CMGNN.

This work mainly focuses on the message-passing mechanism in existing HTGNNs under the semi-supervised setting. Thus, this paper does not analyze the other designs in HTGNNs, such as spectral- and graph-transform-based methods. Theoretical analyses and proofs are based on the CSBM-H with corresponding assumptions about node features, degrees and edges.

## Acknowledgments and Disclosure of Funding

This work is supported by the National Natural Science Foundation of China (Grant No. 62476245), Zhejiang Provincial Natural Science Foundation of China (Grant No. LTGG23F030005).

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

# A  Related Works

**Homophilic Graph Neural Networks**. Graph Neural Networks (GNNs) have showcased impressive capabilities in handling graph-structured data [36, 37]. Traditional GNNs are predominantly founded on the assumption of homophily, broadly categorized into two classes: spectral-based GNNs and spatial-based GNNs. *Firstly*, spectral-based GNNs acquire node representations through graph convolution operations employing diverse graph filters [1, 38, 39]. *Secondly*, spatial-based methods gather information from neighbors and update the representation of central nodes through the message-passing mechanism [27, 29, 28]. Moreover, for a more *comprehensive understanding of existing homophilic GNNs*, several unified frameworks [40, 41] have been proposed. Ma et al. [40] propose that the aggregation process in some representative homophilic GNNs can be regarded as solving a graph denoising problem with a smoothness assumption. Zhu et al. [41] establishes a connection between various message-passing mechanisms and a unified optimization problem. However, these methods have limitations, as the aggregated representations may lose discriminability when heterophilic neighbors dominate [11, 16].

**Heterophilic Graph Neural Networks**. Recently, some heterophilic GNNs have emerged to tackle the heterophily problem [10–13, 33, 16, 42, 43, 17, 22, 23, 44, 32, 35, 34]. *Firstly*, a commonly adopted strategy involves *expanding the neighborhood with higher homophily or richer messages*, such as high order neighborhooods [16, 42], feature-similarity-based neighborhoods [42, 43], and custom-defined neighborhoods [17, 44]. *Secondly*, some approaches [11, 33, 12, 22, 32] aim to *leverage information from heterophilic neighbors*, considering that not all heterophily is detrimental [6]. *Thirdly*, some methods [16, 10, 23, 13] adapt to heterophily by extending the combine function in message passing, creating variations for addition and concatenation.

**Reviewing Heterophilic Graph Neural Networks**. Heterophilic GNNs have attracted more and more research attention. Some surveys have provided a macroscopic view for reviewing heterophilic GNNs, categorizing heterophilic GNNs with shallow analysis. Specifically, Zheng et al. [8] categorizes the designs of heterophilic GNNs into non-local neighbor extensions and GNN architecture refinement. Zhu et al. [9] examines the impact of heterophilic graph characteristics on GNNs. For categorizations, it simply lists some effective designs in heterophilic GNNs. Gong et al. [45] reviews heterophilic graph learning, where message passing is only a minor aspect of its taxonomy with a broader view.

**The Connection Between Message Passing and Heterophily**. Recently, some efforts [6, 9, 14, 15, 46] have begun to investigate the connection between message passing and heterophily. Zhu et al. [9] highlighted that node degrees and compatibility matrices are key factors for message passing under heterophily. In ACM [12], the authors found that the performance curves of the VMP mechanism under different levels of homophily are U-shaped. Similarly, Ma et al. [6] proposed the existence of a special case of heterophily, named "good" heterophily, where the homophily ratios remain low, but the VMP mechanism can achieve strong performance. Luan et al. [15] shows that the low-pass filter works better at very low and very high homophily intervals, while the high-pass filter works better at the low to medium homophily interval. However, these works are limited in studying what kind of data is suitable for the message-passing mechanism, lacking a model-perspective understanding of heterophilic message passing.

# B  Proof of Theorem 3.2

To prove Theorem 3.2, we first calculate the $\text{CMD}(\mathbf{M}_{\text{CSBM-H}})$ and $\text{ENKL}(\mathbf{H}^{\text{LP}})$, respectively.

According to the definitions in CSBM-H, the corresponding compatibility matrix is as follows:

$$\mathbf{M}_{\text{CSBM-H}} = \begin{bmatrix} h, & 1-h \\ 1-h, & h \end{bmatrix}. \tag{18}$$

Thus, the corresponding CMD can be calculated by Eq.5:

$$\text{CMD}(\mathbf{M}_{\text{CSBM-H}}) = \frac{2(|h - (1-h)| + |1 - h - h|)}{2} = 2|2h - 1| = 4|h - \frac{1}{2}|. \tag{19}$$

On the other hand, the original $\mathbf{X}$ and LP features $\mathbf{H}^{\text{LP}} = \hat{\mathbf{A}}\mathbf{X}$ are as follows:

$$
\begin{aligned}
i \in \mathcal{C}_0, x_i \sim N(\boldsymbol{\mu}_0, \sigma_0^2 \mathbf{I}); \ h_i^{LP} \sim N(\tilde{\boldsymbol{\mu}}_0, \tilde{\sigma}_0^2 \mathbf{I}), \\
j \in \mathcal{C}_1, x_j \sim N(\boldsymbol{\mu}_1, \sigma_1^2 \mathbf{I}); \ h_j^{LP} \sim N(\tilde{\boldsymbol{\mu}}_1, \tilde{\sigma}_1^2 \mathbf{I}),
\end{aligned}
\tag{20}
$$

where $\tilde{\boldsymbol{\mu}}_0 = h(\boldsymbol{\mu}_0 - \boldsymbol{\mu}_1) + \boldsymbol{\mu}_1, \tilde{\boldsymbol{\mu}}_1 = h(\boldsymbol{\mu}_1 - \boldsymbol{\mu}_0) + \boldsymbol{\mu}_0, \tilde{\sigma}_0^2 = \frac{h(\sigma_0^2 - \sigma_1^2) + \sigma_1^2}{d_0}, \tilde{\sigma}_1^2 = \frac{h(\sigma_1^2 - \sigma_0^2) + \sigma_0^2}{d_1}$.

Considering the formula of ENKL Eq.2, we first calculate $d_{\mathbf{H}^{\mathrm{LP}}}^2$.

$$
\begin{aligned}
d_{\mathbf{X}}^2 &= \|\boldsymbol{\mu}_0 - \boldsymbol{\mu}_1\|_2^2, \\
d_{\mathbf{H}^{\mathrm{LP}}}^2 &= \|\tilde{\boldsymbol{\mu}}_0 - \tilde{\boldsymbol{\mu}}_1\|_2^2 \\
&= \|h(\boldsymbol{\mu}_0 - \boldsymbol{\mu}_1) + \boldsymbol{\mu}_1 - (h(\boldsymbol{\mu}_1 - \boldsymbol{\mu}_0) + \boldsymbol{\mu}_0)\|_2^2 \\
&= \|(2h-1)(\boldsymbol{\mu}_0 - \boldsymbol{\mu}_1)\|_2^2 \\
&= (2h-1)^2 d_{\mathbf{X}}^2.
\end{aligned} \tag{21}
$$

In CSBM-H, $F_h = 2$, we further assume $d_0 = d_1 = d$, that is, ignoring the influence of node degrees. Now we have $\tilde{\sigma}_0^2 = \frac{h(\sigma_0^2 - \sigma_1^2) + \sigma_1^2}{d}$ and $\tilde{\sigma}_1^2 = \frac{h(\sigma_1^2 - \sigma_0^2) + \sigma_0^2}{d}$.

The ENKL of LP feature is as follows:

$$
\begin{aligned}
\mathrm{ENKL}(\mathbf{H}^{\mathrm{LP}}) &= -d_{\mathbf{H}^{\mathrm{LP}}}^2 \left(\frac{1}{4\tilde{\sigma}_1^2} + \frac{1}{4\tilde{\sigma}_0^2}\right) - \frac{2}{4}\left(\frac{\tilde{\sigma}_0^2}{\tilde{\sigma}_1^2} + \frac{\tilde{\sigma}_1^2}{\tilde{\sigma}_0^2} - 2\right) \\
&= 1 - (2h-1)^2 d_{\mathbf{X}}^2 \left(\frac{d}{4[h(\sigma_1^2 - \sigma_0^2) + \sigma_0^2]} + \frac{d}{4[h(\sigma_0^2 - \sigma_1^2) + \sigma_1^2]}\right) \\
&\quad - \frac{1}{2}\left(\frac{h(\sigma_0^2 - \sigma_1^2) + \sigma_1^2}{h(\sigma_1^2 - \sigma_0^2) + \sigma_0^2} + \frac{h(\sigma_1^2 - \sigma_0^2) + \sigma_0^2}{h(\sigma_0^2 - \sigma_1^2) + \sigma_1^2}\right) \\
&= 1 - (2h-1)^2 d_{\mathbf{X}}^2 \cdot \frac{(\sigma_0^2 + \sigma_1^2)d}{4[-(\sigma_0^2 - \sigma_1^2)^2 h^2 + (\sigma_0^2 - \sigma_1^2)^2 h + \sigma_0^2 \sigma_1^2]} \\
&\quad - \frac{1}{2} \cdot \frac{2(\sigma_0^2 - \sigma_1^2)^2 h^2 - 2(\sigma_0^2 - \sigma_1^2)^2 h + \sigma_0^4 + \sigma_1^4}{-(\sigma_0^2 - \sigma_1^2)^2 h^2 + (\sigma_0^2 - \sigma_1^2)^2 h + \sigma_0^2 \sigma_1^2} \\
&= 1 + d_{\mathbf{X}}^2 \cdot d \cdot \frac{(\sigma_0^2 + \sigma_1^2)(h^2 - h + \frac{1}{4})}{(\sigma_0^2 - \sigma_1^2)^2 h^2 - (\sigma_0^2 - \sigma_1^2)^2 h - \sigma_0^2 \sigma_1^2} \\
&\quad + \frac{(\sigma_0^2 - \sigma_1^2)^2 h^2 - (\sigma_0^2 - \sigma_1^2)^2 h + \frac{1}{2}\sigma_0^4 + \frac{1}{2}\sigma_1^4}{(\sigma_0^2 - \sigma_1^2)^2 h^2 - (\sigma_0^2 - \sigma_1^2)^2 h - \sigma_0^2 \sigma_1^2} \\
&= 1 + \frac{d_{\mathbf{X}}^2 \cdot d(\sigma_0^2 + \sigma_1^2)}{(\sigma_0^2 - \sigma_1^2)^2} \cdot \frac{h^2 - h + \frac{1}{4}}{h^2 - h - \frac{\sigma_0^2 \sigma_1^2}{(\sigma_0^2 - \sigma_1^2)^2}} + \frac{h^2 - h + \frac{\sigma_0^4 + \sigma_1^4}{2(\sigma_0^2 - \sigma_1^2)^2}}{h^2 - h - \frac{\sigma_0^2 \sigma_1^2}{(\sigma_0^2 - \sigma_1^2)^2}} \\
&= \frac{d_{\mathbf{X}}^2 \cdot d(\sigma_0^2 + \sigma_1^2)}{(\sigma_0^2 - \sigma_1^2)^2} \cdot \frac{\frac{1}{4} + \frac{\sigma_0^2 \sigma_1^2}{(\sigma_0^2 - \sigma_1^2)^2}}{(h - \frac{1}{2})^2 - \frac{1}{4} - \frac{\sigma_0^2 \sigma_1^2}{(\sigma_0^2 - \sigma_1^2)^2}} + \frac{\frac{\sigma_0^4 + \sigma_1^4}{2(\sigma_0^2 - \sigma_1^2)^2} + \frac{\sigma_0^2 \sigma_1^2}{(\sigma_0^2 - \sigma_1^2)^2}}{(h - \frac{1}{2})^2 - \frac{1}{4} - \frac{\sigma_0^2 \sigma_1^2}{(\sigma_0^2 - \sigma_1^2)^2}} \\
&\quad + \frac{d_{\mathbf{X}}^2 \cdot d(\sigma_0^2 + \sigma_1^2)}{(\sigma_0^2 - \sigma_1^2)^2} + 2 \\
&= \frac{d_{\mathbf{X}}^2 \cdot d(\sigma_0^2 + \sigma_1^2)}{(\sigma_0^2 - \sigma_1^2)^2} \cdot \frac{\frac{(\sigma_0^2 + \sigma_1^2)^2}{4(\sigma_0^2 - \sigma_1^2)^2}}{(h - \frac{1}{2})^2 - \frac{(\sigma_0^2 + \sigma_1^2)^2}{4(\sigma_0^2 - \sigma_1^2)^2}} + \frac{\frac{(\sigma_0^2 + \sigma_1^2)^2}{2(\sigma_0^2 - \sigma_1^2)^2}}{(h - \frac{1}{2})^2 - \frac{(\sigma_0^2 + \sigma_1^2)^2}{4(\sigma_0^2 - \sigma_1^2)^2}} \\
&\quad + \frac{d_{\mathbf{X}}^2 \cdot d(\sigma_0^2 + \sigma_1^2)}{(\sigma_0^2 - \sigma_1^2)^2} + 2 \\
&= \left[\frac{d_{\mathbf{X}}^2 \cdot d(\sigma_0^2 + \sigma_1^2)}{(\sigma_0^2 - \sigma_1^2)^2} + 2\right] \cdot \left[\frac{\frac{(\sigma_0^2 + \sigma_1^2)^2}{4(\sigma_0^2 - \sigma_1^2)^2}}{(h - \frac{1}{2})^2 - \frac{(\sigma_0^2 + \sigma_1^2)^2}{4(\sigma_0^2 - \sigma_1^2)^2}} + 1\right].
\end{aligned} \tag{22}
$$

Since $\boldsymbol{\mu}_0, \boldsymbol{\mu}_1, \sigma_0^2, \sigma_1^2$ and $d$ are fixed parameters in CSBM-H when only considering the change of homophily $h$, we simplify the above formulation by new variables $a, b$.

$$
\begin{aligned}
a &= \left[\frac{d_{\mathbf{X}}^2 \cdot d(\sigma_0^2 + \sigma_1^2)}{(\sigma_0^2 - \sigma_1^2)^2} + 2\right] \in [2, +\infty), \\
b &= \frac{(\sigma_0^2 + \sigma_1^2)^2}{4(\sigma_0^2 - \sigma_1^2)^2} \in [\frac{1}{4}, +\infty).
\end{aligned} \tag{23}
$$

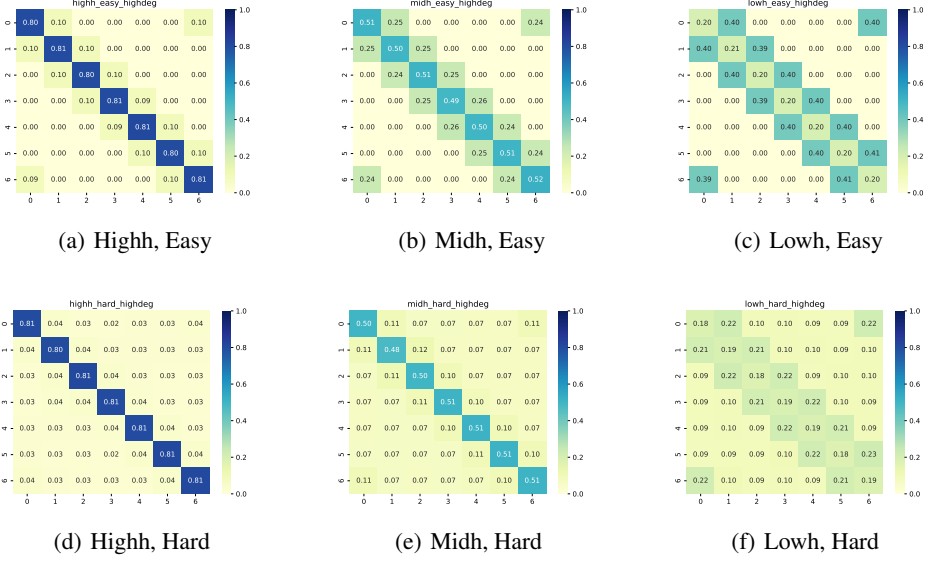

(a) Highh, Easy        (b) Midh, Easy        (c) Lowh, Easy

(d) Highh, Hard        (e) Midh, Hard        (f) Lowh, Hard

Figure 5: The visualization of the compatibility matrix on synthetic graphs.

Thus, the ENKL can be represented as follows:

$$\text{ENKL}(\mathbf{H}^{\text{LP}}) = a\left[\frac{b}{(h - \frac{1}{2})^2 - b} + 1\right]. \tag{24}$$

Furthermore, let $t = |h - \frac{1}{2}| \in [0, \frac{1}{2}]$ since $h \in [0, 1]$, the CMD and ENKL can be represented as follows:

$$\text{CMD}(\mathbf{M}_{\text{CSBM-H}}) = 4t,$$
$$\text{ENKL}(\mathbf{H}^{\text{LP}}) = \frac{ab}{t^2 - b} + a. \tag{25}$$

To simplify the computation, we analyze monotonicity with the aid of derivatives.

$$\frac{d\text{CMD}}{dt} = 4 > 0,$$
$$\frac{d\text{ENKL}}{dt} = \frac{-2abt}{(t^2 - b)^2} \leq 0. \tag{26}$$

That means CMD increases monotonically with t, while ENKL decreases monotonically with t. Thus, the covariance between CMD and ENKL is negative: $\text{Cov}(\text{CMD}(\mathbf{M}_{\text{CSBM-H}}), \text{ENKL}(\mathbf{H}^{\text{LP}})) < 0$.

## C  The Experiments on Synthetic Datasets

To explore the performance impact of homophily level, node degrees, and compatibility matrix (CMs) on simple GNNs, we conduct some experiments on synthetic datasets.

### C.1  Synthetic Datasets

We construct synthetic graphs considering the factors of homophily, CMs, and degrees. For homophily, we set 3 levels including Lowh (0.2), Midh (0.5), and Highh (0.8). For CMs, we set two levels of discriminability, including Easy and Hard. For degrees, we set two levels, including Lowdeg (4) and Highdeg (18). Note that with a certain homophily level, we can only control the non-diagonal elements of CMs. Thus, there are a total of 12 synthetic graphs following the above settings. These synthetic graphs are based on the Cora dataset, which provides node features and labels, which means, only the edges are constructed. We visualize the CMs of these graphs in Figure 5. Since there is no

Table 5: Node classification accuracy of GCN on synthetic datasets.

| Factors | Highh, Easy | Highh, Hard | Midh, Easy | Midh, Hard | Lowh, Easy | Lowh, Hard |
|---------|-------------|-------------|------------|------------|------------|------------|
| **Highd** | 99.15 ± 0.35 | 99.48 ± 0.24 | 86.42 ± 4.13 | 90.52 ± 1.05 | 89.34 ± 2.19 | 39.22 ± 2.34 |
| **Lowd** | 89.98 ± 1.59 | 91.25 ± 0.85 | 70.85 ± 1.59 | 70.20 ± 1.41 | 56.46 ± 2.63 | 40.91 ± 1.75 |

significant difference in CMs between low-degree and high-degree, we only plot the high-degree ones. Further, the edges are randomly constructed under the guidance of these CMs and degrees to form the synthetic graphs.

## C.2 Experiments on Synthetic Datasets

We use GCN to analyze the performance impact of the above factors. The semi-supervised node classification performance of GCN is shown in Table 5 while the baseline performance of MLP (72.54 ± 2.18) is the same among these datasets since their difference is only on edges. From these results, we have some observations: (1) High homophily is not necessary, **GCN can also work well on low homophily but discriminative CM**; (2) Low degrees have a negative impact on performance, especially when the CMs are relatively less discriminative. This also indicates that nodes with lower degrees are more likely to have confused semantic neighborhoods; (3) When handling nodes with confused semantic neighborhoods, GCN may contaminate central nodes with their neighborhoods' messages, which leads to worse performance than MLP. This once again reminds us of the importance of enhancing the CM discriminability.

## D The Posterior Evaluation about the Weighted-CM and CMD in HTGNNs

In this part, we give the details of the empirical posterior evaluation on GloGNN [22], GPR-GNN [23], and ACM-GCN [12].

**GloGNN.** GloGNN learns a global pair-wise coefficient matrix $\mathbf{Z}$ and utilizes it as the aggregation weights during message passing. Thus, we directly calculate this matrix as the practical aggregate weights matrix $\dot{\mathbf{A}}^{glo} = \mathbf{Z}$, then regard $\dot{\mathbf{A}}$ as the neighborhood and calculate the Weighted-CM and its corresponding CMD.

**ACM-GCN.** ACM-GCN merges messages from various filters using adaptive weights, effectively altering edge weights to construct the Weight-CM with optional negative elements due to the high-pass filter. Thus, we leverage the learned weights to rebuild a practical aggregate weights matrix $\dot{\mathbf{A}}^{acm}$ based on the low-pass filter $\hat{\mathbf{A}}$ and high-pass filter $\mathbf{I} - \hat{\mathbf{A}}$, then regard $\dot{\mathbf{A}}^{acm}$ as the neighborhood and calculate the Weighted-CM and its corresponding CMD.

**GPRGNN.** GPR-GNN integrates the CMs of multiple-order neighborhoods with adaptive weights to form a more discriminative Weighted-CM. Thus, we utilize the learned weights to rebuild a practical aggregate weights matrix $\dot{\mathbf{A}}^{gpr}$ based on the multi-hop adjacency matrices $[\mathbf{I}, \mathbf{A}, \mathbf{A}^2, ..., \mathbf{A}^k]$, then regard $\dot{\mathbf{A}}^{gpr}$ as the neighborhood and calculate the Weighted-CM and its corresponding CMD.

The visualization of Weighted-CMs and corresponding CMD on various datasets can be seen in Figure 6.

## E Additional Detailed Implementation of CMGNN

**Considering the influence of node degree in compatibility matrix estimation.** As mentioned in Section 3, the semantic neighborhood of low-degree nodes may display inconsistencies with CM. Thus, nodes with low degrees deserve low weights during the CM estimation. We manually set up a weighting function range in $[0, 1]$:

$$w_i^d = \begin{cases} d_i/2K, & d_i \leq K, \\ 0.25 + d_i/4K, & K < d_i \leq 3K, \\ 1, & otherwise. \end{cases} \quad (27)$$

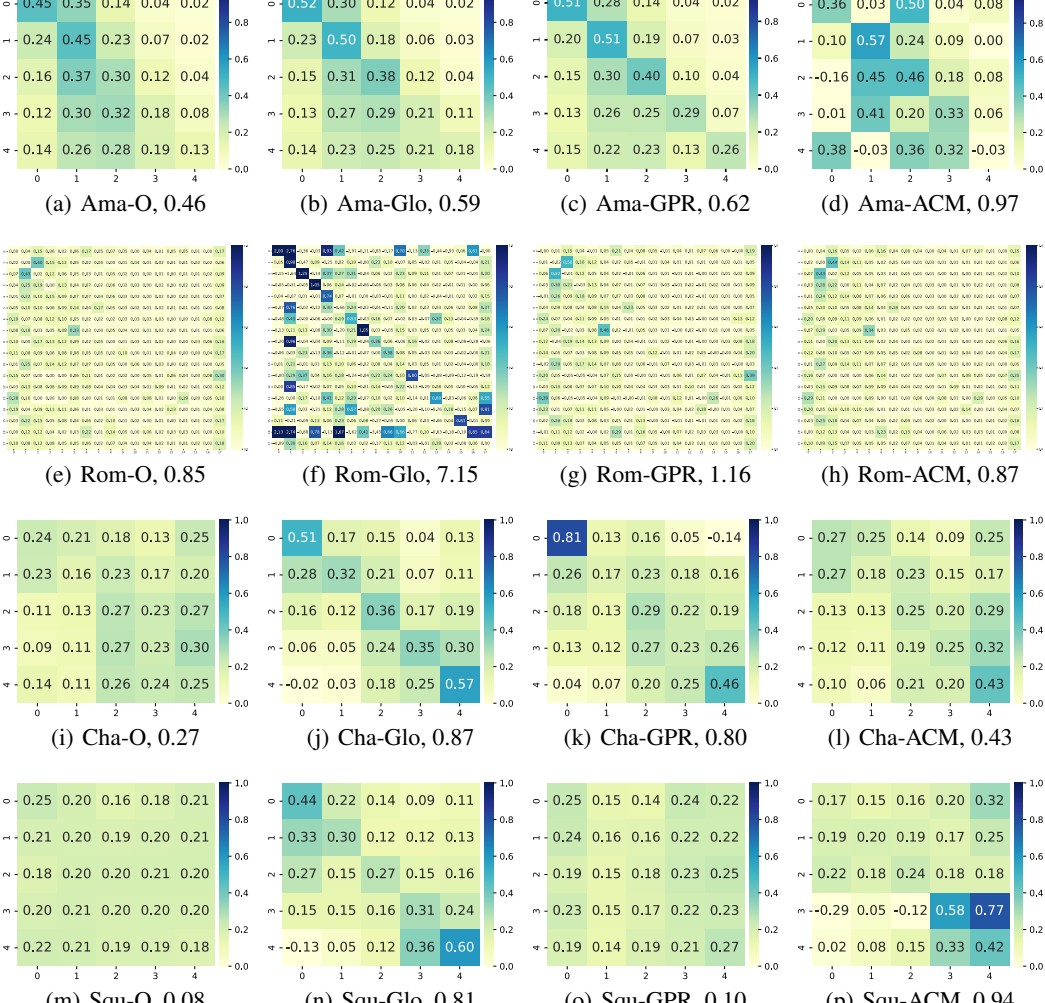

Figure 6: Visualizations of Weighted-CMs and CMD of GloGNN, GPR-GNN, and ACM-GCN on various datasets.

For low-degree nodes, increases in degree should yield more significant benefits compared to high-degree nodes. Beyond a certain threshold, increases in degree yield tiny benefits. We have empirically chosen $K$ and $3K$ as fixed thresholds for the weighting function to simplify the design without multiple attempts. This approach is straightforward and can be substituted with other forms that meet the same criteria. In practice, the compatibility matrix is estimated considering the various node degrees:

$$\hat{\mathbf{M}} = \text{Norm}((\mathbf{w}^d \cdot \mathbf{g} \cdot \hat{\mathbf{C}})^T)\mathbf{C}^{nb}. \tag{28}$$

**The utilization of additional structural features.** In line with existing methods [12, 22], we treat the topology structure as additional node features. These features, shown as the adjacency matrix $\mathbf{A}$, depict node connections. Each row $\mathbf{A}_i$ can be viewed as an extra $N$-dimensional feature of node $i$. Thus, the input representation of the first layer can be obtained in two ways:

$$\mathbf{Z}^0 = [\mathbf{X}\mathbf{W}^X \| \hat{\mathbf{A}}\mathbf{W}^A]\mathbf{W}^0, \text{ or } \mathbf{Z}^0 = \mathbf{X}\mathbf{W}^0. \tag{29}$$

Specifically, (i) using additional features, where $\mathbf{W}^X \in \mathbb{R}^{d_f \times d_r}$, $\mathbf{W}^A \in \mathbb{R}^{N \times d_r}$ and $\mathbf{W}_0 \in \mathbb{R}^{2d_r \times d_r}$ are learnable matrices; (ii) using only attribute features, where $\mathbf{W}^0 \in \mathbb{R}^{d_f \times d_r}$, where $d_r$ is the dimension of node embeddings.

**Message COMBNIE with Adaptive Weights.** The aggregated messages from node ego, raw, and supplementary neighborhoods are $\mathbf{Z}^l_{ego}$, $\mathbf{Z}^l_{raw}$ and $\mathbf{Z}^l_{sup}$, respectively. The combination weights are

---

**Algorithm 1** Algorithm of CMGNN

---

**Require:** Graph $\mathcal{G} = (\mathcal{V}, \mathcal{E}, \mathbf{X}, \mathbf{A}, \mathbf{Y})$, loss weight $\lambda$, epoch $E$
**Ensure:** Predicted labels $\hat{\mathbf{Y}}$
1: Initialize the soft predicted labels $\tilde{C}$ with other elements $\frac{1}{K}$.
2: Construct class prototypes as additional virtual neighbors for all nodes via Eq.9.
3: **for** iteration 1, 2, ..., $E$ **do**
4:    Obtain the input representations for the first layer via Eq.29.
5:    Estimate the compatibility matrix via Eq.10, Eq.11, Eq.27, and Eq.28.
6:    Obtain the output representations through the CM-aware message-passing mechanism via Eq.13, Eq.14, and Eq.15.
7:    Obtain the predicted logits (soft label) $\tilde{\mathbf{C}}$ via Eq.16.
8:    Calculate loss $\mathcal{L}$ via Eq.17.
9:    Back-propagation $\mathcal{L}$ to optimize the weights of networks.
10:    **if** the performance in the validation set improved **then**
11:       update the compatibility matrix with current soft predicted label $\tilde{\mathbf{C}}$.
12:    **end if**
13: **end for**
14: Obtain the predicted labels $\hat{\mathbf{Y}}$ via $\hat{\mathbf{Y}} = \text{Softmax}(\tilde{\mathbf{C}})$.
15: **output** $\hat{\mathbf{Y}}$

---

learned by an MLP with Softmax:

$$[\alpha_{ego}^l, \alpha_{raw}^l, \alpha_{sup}^l] = \text{Softmax}(\text{Sigmoid}([\mathbf{Z}_{ego}^l \| \mathbf{Z}_{raw}^l \| \mathbf{Z}_{sup}^l \| \mathbf{d}] \mathbf{W}_{att}^l) \mathbf{W}_{mix}^l), \qquad (30)$$

where $\mathbf{W}_{att}^l \in \mathbb{R}^{(3d_r+1) \times 3}$ and $\mathbf{W}_{mix}^l \in \mathbb{R}^{3 \times 3}$ are two learnable weight matrixes, $\mathbf{d}$ is the node degrees which may be helpful to weights learning.

**The Message Passing of Class Prototypes.** Specifically, the virtual prototype nodes are viewed as additional nodes, which have the same message-passing mechanism as real nodes:

$$\mathbf{Z}^{ptt,l} = \text{diag}(\alpha_{ego}^{ptt,l}) \mathbf{Z}^{ptt,l-} \mathbf{W}_{ego}^l + \text{diag}(\alpha_{raw}^{ptt,l}) \hat{\mathbf{A}}^{ptt} \mathbf{Z}^{ptt,l-1} \mathbf{W}_{raw}^l$$
$$+ \text{diag}(\alpha_{sup}^{ptt,l})(\mathbf{A}^{ptt,sup} \odot \mathbf{B}^{ptt,sup}) \mathbf{Z}^{ptt,l-1} \mathbf{W}_{sup}^l, \qquad (31)$$
$$\mathbf{Z}^{ptt} = \mathop{\big\|}_{l=0}^{L} \mathbf{Z}^{ptt,l},$$

where $\mathbf{A}^{sup,ptt} = \mathbf{1} \in \mathbb{R}^{K \times K}$ and $\mathbf{B}^{sup,ptt} = \hat{\mathbf{C}}^{ptt} \hat{\mathbf{M}}$ are similar with those of real nodes.

## F    More Details about the Experimental Settings

In this section, we describe the details of the new benchmarks, including (i) the reason why we need a new benchmark: drawbacks of existing datasets; (ii) detailed descriptions of newly organized datasets; (iii) baseline methods and the codebase; (iv) details of obtaining benchmark performance; and (v) detailed experimental settings of CMGNN.

### F.1    Drawbacks in Existing Datasets

Existing works mostly follow the settings and datasets used in Pei et al. [17], including 6 heterophilic datasets (Cornell, Texas, Wisconsin, Actor, Chameleon, and Squirrel) and 3 homophilic datasets (Cora, Citeseer, and Pubmed). Platonov et al. [26] pointed out serious data leakages in Chameleon and Squirrel, while Cornell, Texas, and Wisconsin are too small with very imbalanced classes. Further, we revisit other datasets and discover new drawbacks: (i) In the ten splits of Citeseer, there are two inconsistent ones, which have smaller training, validation, and test sets that could cause issues with statistical results; (ii) Cora's data split ratios are inconsistent with the expected ones. These drawbacks may lead to certain issues in the conclusions of previous works.

Therefore, to build a comprehensive and fair benchmark for model effectiveness evaluation, we newly organize 13 datasets with unified splitting across various homophily values and scales.

## F.2 Newly Organized Datasets

In our benchmark, we adopt ten different types of publicly available datasets with a unified splitting setting (48%/32%/20% for training/validation/testing) for fair model comparison, including **Roman-Empire** [26], **Amazon-Ratings** [26], **Chameleon-F** [26], **Squirrel-F** [26], **Actor** [17], **Flickr** [47], **BlogCatalog** [47], **Pubmed** [48], **Penn94** [49], **Twitch-Gamer** [49], **Genius** [49], **Pokec** [49] and **Snap-Patents** [49]. The datasets have a variety of homophily values from low to high. The statistics and splitting of these datasets are shown in Table 6. The detailed description of the datasets is as follows:

Table 6: Statistics and splitting of the experimental benchmark datasets.

| Dataset | Nodes | Edges | Attributes | Classes | Avg. Degree | Undirected | Homophily | Train / Valid / Test |
|---|---|---|---|---|---|---|---|---|
| Roman-Empire | 22,662 | 65,854 | 300 | 18 | 2.9 | ✓ | 0.05 | 10,877 / 7,251 / 4,534 |
| Amazon-Ratings | 24,492 | 186,100 | 300 | 5 | 7.6 | ✓ | 0.38 | 11,756 / 7,837 / 4,899 |
| Chameleon-F | 890 | 13,584 | 2,325 | 5 | 15.3 | ✗ | 0.25 | 427 / 284 / 179 |
| Squirrel-F | 2,223 | 65,718 | 2,089 | 5 | 29.6 | ✗ | 0.22 | 1,067 / 711 / 445 |
| Actor | 7,600 | 30,019 | 932 | 5 | 3.9 | ✗ | 0.22 | 3,648 / 2,432 / 1,520 |
| Flickr | 7,575 | 479,476 | 12,047 | 9 | 63.3 | ✓ | 0.24 | 3,636 / 2,424 / 1,515 |
| BlogCatalog | 5,196 | 343,486 | 8,189 | 6 | 66.1 | ✓ | 0.40 | 2,494 / 1,662 / 1,040 |
| Pubmed | 19,717 | 88,651 | 500 | 3 | 4.5 | ✓ | 0.80 | 9,463 / 6,310 / 3,944 |
| Penn94 | 41,554 | 1,362,229 | 5 | 2 | 65.6 | ✗ | 0.47 | 19,945 / 13,297 / 8,312 |
| Twitch-Gamer | 168,114 | 6,797,557 | 7 | 2 | 40.4 | ✗ | 0.55 | 80,694 / 53,796 / 33,624 |
| Genius | 421,961 | 984,979 | 12 | 2 | 2.3 | ✗ | 0.62 | 202,541 / 135,027 / 84,393 |
| Pokec | 1,632,803 | 30,622,564 | 65 | 2 | 18.8 | ✗ | 0.45 | 783,667 / 522,444 / 326,529 |
| Snap-Patents | 2,923,922 | 13,975,788 | 269 | 5 | 4.8 | ✗ | 0.07 | 1,403,482 / 935,655 / 584,785 |

- **Roman-Empire**[2] [26] is derived from the extensive article on the Roman Empire found on the English Wikipedia, chosen for its status as one of the most comprehensive entries on the platform. It contains 22,662 nodes and 65,854 edges between nodes. Each node represents an individual word from the text, with the total number of nodes mirroring the length of the article. An edge between two nodes is established under one of two conditions: the words are sequential in the text or they are linked in the sentence's dependency tree, indicating a grammatical relationship where one word is syntactically dependent on the other. Consequently, the graph is structured as a chain graph, enriched with additional edges that represent these syntactic dependencies. It encompasses a total of 18 distinct node classes, with each node being equipped with 300-dimensional attributes obtained by fastText word embeddings [50].

- **Amazon-Ratings**[2] [26] is sourced from the Amazon product co-purchasing network metadata dataset [51]. It contains 24,492 nodes and 186,100 edges between nodes. The nodes within this graph represent products, encompassing a variety of categories such as books, music CDs, DVDs, and VHS video tapes. An edge between nodes signifies that the respective products are often purchased together. The objective is to forecast the average rating assigned to a product by reviewers, with the ratings being categorized into five distinct classes. For the purpose of node feature representation, we have utilized the 300-dimensional mean values derived from fastText word embeddings [50], extracted from the textual descriptions of the products.

- **Chameleon-F** and **Squirrel-F**[2] [26] are specialized collections of Wikipedia page-to-page networks [52], of which the data leakage nodes are filtered out by Platonov et al. [26]. Within these datasets, each node symbolizes a web page, and edges denote the mutual hyperlinks that connect them. The node features are derived from a selection of informative nouns extracted directly from Wikipedia articles. For the purpose of classification, nodes are categorized into five distinct groups based on the average monthly web traffic they receive. Specifically, Chameleon-F contains 890 nodes and 13,584 edges between nodes, with each node being equipped with 2,325-dimensional features. Squirrel-F contains 2,223 nodes and 65,718 edges between nodes, with each node being equipped with a 2,089-dimensional feature vector.

- **Actor**[3] [17] is an actor-centric induced subgraph derived from the broader film-director-actor-writer network, as originally presented by Tang et al. [53]. In this refined network, each node corresponds to an individual actor, and the edges signify the co-occurrence of these actors on the same Wikipedia page. The node features are identified through the presence of certain keywords found within

---

[2]`https://github.com/yandex-research/heterophilous-graphs/tree/main/data`
[3]`https://github.com/bingzhewei/geom-gcn/tree/master/new_data/film`

the actors' Wikipedia entries. For the purpose of classification, the actors are organized into five distinct categories based on the words of the actor's Wikipedia. Statistically, it contains 7,600 nodes and 30,019 edges between nodes, with each node being equipped with a 932-dimensional feature vector.

- **Flickr** and **Blogcatalog**[4] [47] are two datasets of social networks, originating from the blog-sharing platform BlogCatalog and the photo-sharing platform Flickr, respectively. Within these datasets, nodes symbolize the individual users of the platforms, while links signify the followship relations that exist between them. In the context of social networks, users frequently create personalized content, such as publishing blog posts or uploading and sharing photos with accompanying tag descriptions. These textual contents are consequently treated as attributes associated with each node. The classification objective is to predict the interest group of each user. Specifically, Flickr contains 7,575 nodes and 479,476 edges between nodes. The graph encompasses a total of 9 distinct node classes, with each node being equipped with a 12047-dimensional attribute vector. BlogCatalog contains 5,196 nodes and 343,486 edges between nodes. The graph encompasses a total of 6 distinct node classes, with each node being equipped with 8189-dimensional attributes.

- **Pubmed**[5] [48] is a classical citation network consisting of 19,717 scientific publications with 44,338 links between them. The text contents of each publication are treated as its node attributes, and thus each node is assigned a 500-dimensional attribute vector. The target is to predict which of the paper categories each node belongs to, with a total of 3 candidate classes.

- **Penn94**[6] [49] is a friendship network derived from the Facebook 100 networks, featuring university students from 2005 [54]. In this network, each node represents a student and is labeled with the user's reported gender. The node features include major, second major or minor, dorm or house, year, and high school.

- **Twitch-Gamer**[6] [49] is a subgraph from the streaming platform Twitch, with nodes representing users and edges connecting mutual followers [55]. Node features encompass the number of views, creation and update dates, language, lifetime, and account status. The task is to predict whether a channel contains explicit content.

- **Genius**[6] [49] is a subnetwork extracted from genius.com, a website for crowdsourced annotations of song lyrics [56]. In this graph, nodes represent users, and edges connect users who follow each other. User features include expertise scores, contribution counts, roles, and more. Some users are labeled as "gone", indicating a higher likelihood of being spam accounts. Our goal is to predict whether a user is marked as "gone".

- **Pokec**[6] [49] is a friendship graph from a Slovak online social network, with nodes representing users and edges indicating directed friendship relations [57]. Node features are derived from profile information, such as geographical region, registration time, and age. The task is to classify users based on their gender.

- **Snap-Patents**[6] [49] is a U.S. patent network, where nodes correspond to patents and edges denote citation relationships [58]. Node features are derived from patent metadata. The task is to classify patents into five categories based on the time of their grant.

### F.3   Baseline Methods and the Codebase

For comprehensive comparisons, we choose 20 representative baseline methods as in the benchmark, which can be categorized into four main groups of works as follows:

(i) **Shallow Model**: MLP;

(ii) **Homopihlous Graph Neural Networks**: GCN [1], GAT [27], GraphSAGE [28], APPNP [29], and GCNII [30];

(iii) **Heterophilous Graph Neural Networks**: H2GCN [16], MixHop [10], GBK-GNN [31], GGCN [32], GloGNN [22], HOGGCN [33], GPR-GNN [23], ACM-GCN [12], OrderedGNN [13], M2MGNN [45], and $N^2$ [34];

(iv) **Compatibility Matrix-based Models**: CLP [25], EPFGNN [24], and CPGNN [7].

---

[4] https://github.com/TrustAGI-Lab/CoLA/tree/main/raw_dataset
[5] https://linqs.soe.ucsc.edu/datac
[6] https://github.com/CUAI/Non-Homophily-Large-Scale/tree/master/data

To explore the performance of baseline methods on newly organized datasets and facilitate future expansions, we collect the official/reproduced codes from GitHub and integrate them into a unified codebase. Specifically, all methods share the same data loaders and evaluation metrics. One can easily run different methods with only parameters changing within the codebase. The codebase is based on the widely used PyTorch[7] framework, supporting both DGL[8] and PyG[9]. Detailed usage of the codebase is available in the Readme file of the codebase.

### F.4 Details of Obtaining Benchmark Performance

Following the settings in existing methods, we construct 10 random splits (48%/32%/20% for train/valid/test) for each dataset and report the average performance among 10 runs on them, along with the standard deviation. For all baseline methods except MLP, GCN, and GAT, we conduct parameter searches within the search space recommended by the original papers. The searches are based on the NNI framework with an annealing strategy. We use Adam as the optimizer for all methods. Each method has dozens of search trails according to their time costs, and the best performances are reported. The currently known optimal parameters of each method are listed in the codebase. We run these experiments on NVIDIA GeForce RTX 3090 GPUs with 24G memory. The out-of-memory error during model training is reported as OOM in Table 2 and 3.

### F.5 Detailed Experimental Settings of CMGNN

CMGNN has the same experimental settings within the benchmark, including datasets, splits, evaluations, hardware, optimizer, and so on.

**Parameters Search Space.** We list the search space of parameters in Table 7, where patience is for the maximum epoch early stopping, n_hidden is the embedding dimension of hidden layers as well as the representation dimension $d_r$, relu_varient decides ReLU applying before message aggregation or not as in Luan et al. [12], structure_info determines whether to use structure information as supplement node features or not.

Table 7: Parameter search space of our method.

| Parameters | Range |
|---|---|
| learning rate | {0.001, 0.005, 0.01, 0.05} |
| weight_decay | {0, 1e-7, 5e-7, 1e-6, 5e-6, 5e-5, 5e-4} |
| patience | {200, 400} |
| dropout | [0, 0.9] |
| $\lambda$ | {0, 0.01, 0.1, 1, 10} |
| layers | {1, 2, 4, 8} |
| n_hidden | {32, 64, 128, 256} |
| relu_variant | {True, False} |
| structure_info | {True, False} |

## G More Details about Experiments

In this section, we show some additional experimental results and analyses.

### G.1 Detailed Analysis about the comparison between CMGNN and existing CM-based methods

Specifically, CMGNN achieves better performance and benefits from the approach of utilizing CM in the following aspects: (i) Better robustness for low-quality pseudo labels: Existing CM-based methods utilize CM to guide the weights of propagation, which can lead to error accumulation with

---

[7]https://pytorch.org
[8]https://www.dgl.ai
[9]https://www.pyg.org

Table 8: Ablation study results (%) on the effects of additional structural features, where True denotes CMGNN with additional structural features and False denotes CMGNN with only node features.

| Structural Features | Roman-Empire | Amazon-Ratings | Chameleon-F | Squirrel-F | Actor | Flickr | BlogCatalog | Pubmed |
|---|---|---|---|---|---|---|---|---|
| True | 68.43 ± 2.23 | **52.13 ± 0.55** | **45.70 ± 4.92** | **41.89 ± 2.34** | 35.72 ± 0.75 | **92.66 ± 0.46** | 96.47 ± 0.58 | 88.90 ± 0.45 |
| False | **84.35 ± 1.27** | 51.41 ± 0.57 | 44.85 ± 5.64 | 40.49 ± 1.55 | **36.82 ± 0.78** | 92.05 ± 0.75 | **97.00 ± 0.52** | **89.99 ± 0.32** |

Table 9: Node classification accuracy comparison (%) among nodes with different degrees.

| Dataset | Amazon-Ratings | | | | | Flickr | | | | | BlogCatalog | | | | |
|---|---|---|---|---|---|---|---|---|---|---|---|---|---|---|---|
| Deg. Prop.(%) | 0~20 | 20~40 | 40~60 | 60~80 | 80~100 | 0~20 | 20~40 | 40~60 | 60~80 | 80~100 | 0~20 | 20~40 | 40~60 | 60~80 | 80~100 |
| **CMGNN** | **59.78** | **58.36** | **53.08** | 41.74 | 47.86 | **92.56** | **91.19** | 92.71 | **93.24** | 93.65 | **94.13** | **97.17** | **98.29** | **97.99** | **97.47** |
| ACM-GCN | 57.35 | 56.21 | 51.74 | 41.55 | 46.47 | 90.44 | 91.17 | **92.85** | **93.19** | 89.50 | 92.17 | 96.68 | 97.83 | 97.84 | 96.51 |
| OrderedGNN | 56.32 | 56.16 | 51.20 | **41.85** | **50.26** | 86.48 | 90.07 | 92.40 | 92.79 | 93.40 | 92.19 | 96.09 | 97.48 | 97.36 | 96.27 |
| GCNII | 50.61 | 49.94 | 47.49 | **41.85** | 47.76 | 87.49 | 90.54 | 92.29 | 92.68 | **95.09** | 92.81 | 96.73 | 97.58 | 97.90 | 97.43 |

| Dataset | Roman-Empire | | | | | Chameleon-F | | | | | Actor | | | | |
|---|---|---|---|---|---|---|---|---|---|---|---|---|---|---|---|
| Deg. Prop.(%) | 0~20 | 20~40 | 40~60 | 60~80 | 80~100 | 0~20 | 20~40 | 40~60 | 60~80 | 80~100 | 0~20 | 20~40 | 40~60 | 60~80 | 80~100 |
| **CMGNN** | **88.60** | **87.00** | **85.59** | **86.25** | **74.33** | 40.73 | **45.28** | **56.02** | **46.64** | 39.93 | 35.56 | 37.14 | 38.40 | 36.03 | 36.84 |
| ACM-GCN | 79.00 | 77.87 | 73.52 | 72.09 | 53.77 | 39.51 | 41.21 | 52.25 | 45.80 | **47.09** | 34.48 | 36.58 | 36.27 | 34.63 | **37.46** |
| OrderedGNN | **88.60** | **87.00** | 85.56 | 84.68 | 69.69 | **43.21** | 44.51 | 49.16 | 38.27 | 32.23 | 35.94 | 38.06 | 37.87 | 35.77 | 37.15 |
| GCNII | 86.79 | 85.14 | 85.20 | 84.75 | 71.09 | 34.84 | 42.56 | 47.50 | 40.45 | 41.84 | **36.89** | 37.20 | **38.53** | **38.02** | 36.99 |

| Dataset | Squirrel | | | | | Pubmed | | | | | | | | | |
|---|---|---|---|---|---|---|---|---|---|---|---|---|---|---|---|
| Deg. Prop.(%) | 0~20 | 20~40 | 40~60 | 60~80 | 80~100 | 0~20 | 20~40 | 40~60 | 60~80 | 80~100 | | | | | |
| **CMGNN** | **45.37** | **47.10** | **45.25** | **34.86** | 37.10 | 89.32 | 89.33 | 89.31 | **92.62** | **89.39** | | | | | |
| ACM-GCN | 41.12 | 44.30 | 44.22 | 32.97 | **42.10** | 89.60 | **89.54** | 89.58 | 92.02 | 89.23 | | | | | |
| OrderedGNN | 43.78 | 45.53 | 43.09 | 27.90 | 28.48 | 89.67 | 89.37 | 89.45 | 92.54 | 89.02 | | | | | |
| GCNII | 43.08 | 45.55 | 43.65 | 33.07 | 38.05 | **89.77** | 89.50 | 89.24 | 92.45 | 88.86 | | | | | |

inaccurate pseudo labels. This is a common limitation of CM-based methods. In CMGNN, the CM is used to construct desired messages while original neighborhoods are preserved, mitigating the impact of inaccurate pseudo labels. (ii) Unlock the effectiveness of CM for low-degree nodes: Existing CM-based methods redefine pair-wise relations only for existing edges, limiting the effectiveness of CMs for low-degree nodes. In CMGNN, virtual neighbors can provide prototype messages from every class, enhancing neighborhood messages for low-degree or even isolated nodes. (iii) More accurate estimation of CM: While existing CM-based methods take naive approaches to estimate or initialize CM, CMGNN considers the effects of node degrees and model prediction confidence, resulting in more accurate CM estimation, especially in real-world situations. Additionally, CM in CMGNN is continuously updated with more accurate pseudo labels, creating a positive cycle.

## G.2 Ablation study on additional structural features

Utilizing additional structural features is a common approach in heterophilous GNNs that offers another way to use connection relationships, introducing both discriminant and redundant information. Thus, it presents a trade-off between the advantages and disadvantages. We conducted an ablation study to examine its effects and report the results in Table 8. The additional structure features have positive effects on four datasets, while others have negative effects. It doesn't significantly impact performance except for Roman-Empire. Moreover, CMGNN can still achieve competitive results without using additional structural features.

## G.3 Performance on Nodes with Various Levels of Degrees

To verify the effect of CMGNN on low-degree nodes, we divide the test set nodes into 5 parts according to their degrees and report the classification accuracy respectively. We compare CMGNN with 3 top-performance methods and show the results in Table 9. In general, nodes with low degrees tend to have incomplete and noisy semantic neighborhoods. Thus, our outstanding performances on the top 20% nodes with the least degree demonstrate the effectiveness of CMGNN for providing supplementary neighborhood messages. Further, we can find that OrderedGNN and GCNII are good at dealing with nodes with high degrees, while ACM-GCN is relatively good at nodes with low degrees. And CMGNN , to a certain extent, can be adapted to both situations at the same time.

Table 10: Performance comparison on low-label rate setting.

| Method | Roman-Empire | Amazon-Ratings | Chameleon-F | Pubmed |
|---|---|---|---|---|
| MLP | 58.14±1.40 | 37.03±0.43 | 36.74±2.97 | 83.07±0.55 |
| GCN | 33.53±0.77 | 37.32±0.52 | 36.91±2.22 | 83.95±0.62 |
| GCNII | 63.27±0.72 | 39.93±0.87 | 38.40±3.27 | **86.04±0.74** |
| ACM-GCN | 59.93±2.03 | 39.96±0.81 | 37.43±2.84 | 85.54±0.79 |
| OrderedGNN | 64.76±2.20 | 40.00±0.90 | 38.91±2.57 | 85.59±0.68 |
| **CMGNN** | **65.93±2.17** | **40.02±0.86** | **40.11±2.82** | 85.62±0.94 |

### G.4 Performance on low label rate setting

We conduct an experiment to investigate the performance of CMGNN under the low label rate setting. The label rate for training is set as 5% to meet the low label rate setting, and the datasets include Roman-Empire, Amazon-Ratings, Chameleon-F, and Pubmed. We compare CMGNN with 2 base and 3 top-performance baseline methods, including MLP, GCN, GCNII, ACMGCN, and OrderedGNN. The classification accuracy comparison is as Table 10. Consistent with the main results, CMGNN can also achieve outstanding performance on the low label rate setting, demonstrating the effectiveness of CMGNN on handling the noise issue.

### G.5 Visualization of Weighted-CM and CMD of CMGNN

We visualize the original and Weighted-CM of CMGNN along with the corresponding CMDs in Figure 7. Obviously, CMGNN has increased the CMDs with Weighted-CMs. This shows that even with incomplete node labels, CMGNN can estimate and enhance high-quality CMs that provide valuable neighborhood information to nodes. Meanwhile, it can adapt to graphs with various levels of heterophily.

### G.6 Efficiency Study

**Complexity Analysis.** The number of learnable parameters in layer $l$ of CMGNN is $3d_r(d_r + 1) + 9$, compared to $d_r d_r$ in GCN and $3d_r(d_r + 1) + 9$ in ACM-GCN, where $d_r$ is the dimension of representations. The time complexity of layer $l$ is composed of three parts:

(i) AGGREGATE: $O(Nd_r^2)$, $O(Nd_r^2 + Md_r)$ and $O(Nd_r^2 + NKd_r)$ for node ego, raw neighborhood and the sufficient neighborhood respectively, where $N$ and $M = |\mathcal{E}|$ denotes the number of nodes and edges;

(ii) COMBINE: $O(3N(3d_r+1)+12N)$ for calculating adaptive weights and $O(3N)$ for combination;

(iii) Final: $O(1)$ for concatenations.

Thus, the overall time complexity of $L$-layer CMGNN is $O(L(Nd_r(3d_r+K+9)+Md_r+18N)+1)$, or $O(LNd_r^2 + LMd_r)$ for brevity.

**Experimental Running Time.** We report the actual average running time (ms per epoch) of baseline methods and CMGNN in Table 11 for comparison. The results demonstrate that CMGNN can balance both performance effectiveness and running efficiency.

**Trade-off Analysis between Effectiveness and Efficiency**. We have also visualized the trade-off between performance accuracy and empirical runtime compared to baseline methods in Figure 8. From the results, we can see that CMGNN achieves the best performance with relatively low time consumption. Compared with OrderedGNN and GCNII, which have the second- and third-best average ranks, CMGNN offers both better accuracy and lower time consumption.

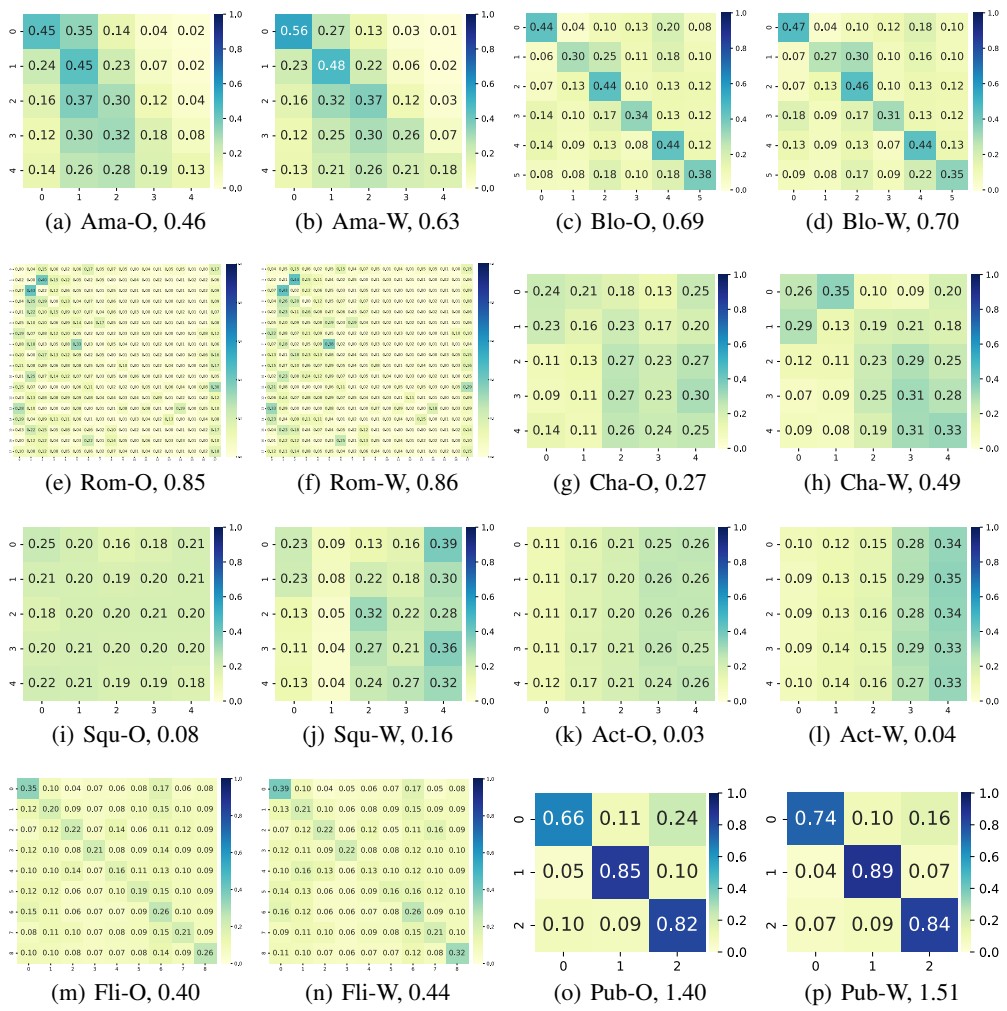

Figure 7: The visualization of original (O) and Weighted-CM (W) of CMGNN along with the CMDs on all small-scale datasets.

Table 11: Efficiency study results of average model running time (ms/epoch). OOM denotes an out-of-memory error during the model training.

| Method | Roman-Empire | Amazon-Ratings | Chameleon-F | Squirrel-F | Actor | Flickr | BlogCatalog | Pubmed |
|---|---|---|---|---|---|---|---|---|
| MLP | 7.8 | 7.0 | 6.1 | 6.5 | 6.3 | 9.1 | 6.7 | 6.1 |
| GCN | 33.8 | 33.4 | 7.9 | 20.6 | 34.4 | 37.2 | 30.4 | 35.6 |
| GAT | 15.9 | 67.3 | 10.3 | 14.0 | 30.8 | 66.2 | 17.6 | 33.4 |
| APPNP | 14.6 | 15.9 | 13.9 | 21.3 | 14.6 | 20.2 | 23.2 | 21.2 |
| GCNII | 29.4 | 28.4 | 37.3 | 19.6 | 37.7 | 84.2 | 97.6 | 258.0 |
| CPGNN | 12.7 | 20.3 | 12.2 | 13.4 | 13.6 | 18.9 | 16.7 | 14.0 |
| H2GCN | 20.0 | 31.2 | 17.2 | 32.4 | 55.6 | 415.7 | 165.5 | 39.0 |
| MixHop | 434.6 | 486.3 | 21.9 | 31.0 | 30.6 | 90.4 | 81.6 | 89.5 |
| GBK-GNN | 119.8 | 191.8 | 31.0 | 238.1 | 157.9 | OOM | OOM | 137.0 |
| GGCN | OOM | OOM | 55.7 | 42.1 | 199.8 | 111.2 | 108.7 | 2290.8 |
| GloGNN | 25.4 | 19.3 | 121.8 | 23.3 | 1292 | 562.9 | 30.9 | 43.2 |
| HOGGCN | OOM | OOM | 25.2 | 54.3 | 1002.9 | 707.3 | 367.4 | OOM |
| GPR-GNN | 15.9 | 12.5 | 22.3 | 23.2 | 16.7 | 15.9 | 14.7 | 13.2 |
| ACM-GCN | 56.7 | 56.7 | 26.1 | 29.7 | 22.5 | 60.7 | 31.7 | 37.1 |
| OrderedGNN | 86.0 | 110.8 | 49.5 | 60.1 | 67.8 | 107.0 | 88.3 | 88.1 |
| M2MGNN | 275.6 | 84.3 | 52.7 | 169.1 | 400.2 | 136.7 | 220.5 | 151.0 |
| N$^2$ | 236.8 | 184.4 | 172.0 | 160.6 | 134.5 | 191.2 | 116.5 | 184.9 |
| **CMGNN** | 51.5 | 93.5 | 62.5 | 64.7 | 19.0 | 52.5 | 69.8 | 102.9 |

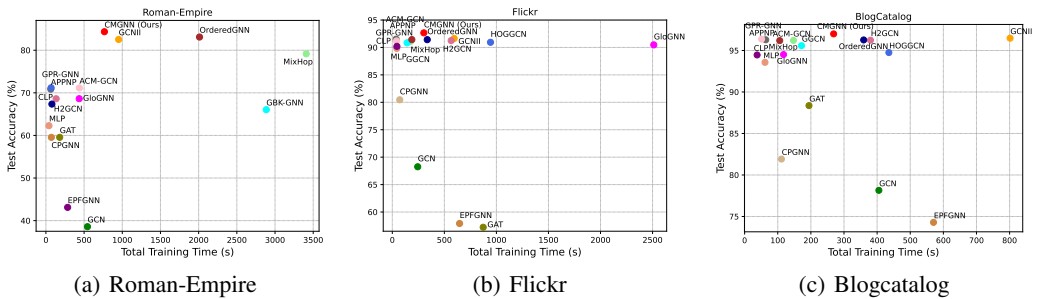

(a) Roman-Empire      (b) Flickr      (c) Blogcatalog

Figure 8: Visualizations of the trade-off between performance accuracy and training time compared with baseline methods on three representative datasets.

