# OpenReview forum: "Understanding and Enhancing Message Passing on Heterophilic Graphs via Compatibility Matrix"
_NeurIPS.cc/2025/Conference — NeurIPS 2025 poster_

### Official Review · Reviewer_ZGc9 · 2025-06-22

**Clarity:** 3
**Significance:** 2
**Originality:** 3
**Rating:** 5
**Confidence:** 4

**Summary:**

The authors introduce a novel Correlation Theorem that rigorously links the separability of node classes to the distinguishability of the underlying compatibility matrix, validating this relationship through synthetic experiments and arguing that the strong empirical performance of heterophilous GNNs stems from enhanced compatibility-matrix distinguishability. They further propose a compatibility-guided message-passing framework that dynamically weights neighbor contributions based on compatibility scores, explicitly designed to mitigate issues arising from low-degree nodes and noisy edges. Finally, they conduct an extensive study on 19 real-world heterophilous benchmarks, demonstrating that the proposed model consistently outperforms existing baselines across both classification accuracy and robustness metrics.

**Questions:**

1. Do you have any sense on the training/optimization dynamics of the compatibility matrix? To me, this is an interesting component since in your formulation it depends on the GNN predictions, so both the GNN and the compatibility matrix are being jointly optimized. Addressing this either theoretically or empirically would be appreciated and interesting.
2. Does the result in Theorem 3.2 generalize to the multiclass case? How do other graph distributions other than the CSBM fit into the theorem? Do you expect the result to hold when the degrees from the classes and across nodes are not equal? This case is particularly important since the method ultimately addresses the low-degree issue for nodes, common in many social networks (powerlaw distributions). Related to weakness #2, how does this analyses build on existing ones either looking at node distinguishability or the class compatibility matrix?
3. It would be interesting to see a detailed investigation of which types of nodes are benefiting over other methods. For example, if there were a way to show that low degree nodes benefit the most over other methods, perhaps by looking at accuracy specifically for low-degree nodes compared to other methods that don't take into account low-degree issues, this would be interesting. Similarly, for graphs with high noise or a low label rate, I would also expect the method to show improvements over other baselines.

**Ethical Concerns:**

["NO or VERY MINOR ethics concerns only"]

**Final Justification:**

Overall, the paper proposed a novel approach to leveraging the compatibility matrix, a powerful tool for homophilic and heterophilic graph learning. While the compatibility matrix has been proposed in the past, specifically addressing low degree nodes with the sufficient neighborhood supplement and addressing uncertainty with confidence based CM estimation, is novel as far as I'm aware. Extensive experiments were conducted to evaluate the utility of the approach. During the rebuttal period, the authors addressed each of my concerns with additional experiments or pointers to existing analyses in the Appendix. Thus, I recommend this paper for acceptance.

**Limitations:**

Yes

**Paper Formatting Concerns:**

No formatting concerns.

**Quality:**

3

**Strengths And Weaknesses:**

**Strengths**
- Specifically targeting low node degree and high noise is very important and many works have outlined issues in these regimes. I liked the use of the $K$ virtual nodes to address the degree issue and the confidence estimation to address the noise issue. The combination of these components fit very well in the overall architecture.
- Experiments were very rigorous with 19 heterophilous benchmarks and impressive performance compared to many of the existing and popular heterophilous models. The ablation analysis was also nice to see.
- There was sufficient discussion against how their approach differed from existing compatibility-based models. This highlighted the contributions of the paper.

**Weaknesses**
- I think the theoretical section is a bit weak. Since Theorem 3.2 considers the two class case, it is really asking the question of if homophily correlates with node distinguishability. The compatibility matrix is interesting in my view since it generalizes homophily from the two-class case to the multi-class case. Since it is well-known that homophily correlates with class distinguishability, this seems to be not a major finding. There are also analyses that have shown the relationship between the class compatibilty matrix and node distinguishability, so I'm not sure how significant the theoretical contributions are.
- I think the work would benefit from a discussion of the similarities and differences with with existing analyses. Throughout the analyses I was unsure of the contributions. Thus making explicit the relations to other analyses and how this analyses builds on existing ones would be beneficial to the reader.

---

> ### Author Rebuttal · Authors · 2025-07-31
>
> We would like to thank you for your comprehensive review. We have carefully considered your comments and suggestions, and the following are our detailed responses.
>
>
> > **Weakness #1 & Question #2: The extension of Theorem 3.2.**
>
> Answer #1: Thank you for this comment! It is indeed important to evaluate the effectiveness of Theorem 3.2 in real-world settings, including multi-class, imbalanced labels, imbalanced degrees, and so on, which are beyond the CSBM-H model. Thus, we conduct experiments to investigate the CMD-ENKL relationship on real datasets, and the results show that **the negative correlation between CMD and ENKL still holds in various real-world settings**.
>
> The details are as follows:
> * **CMD**: The CMDs of fixed CM and Weighted-CM on real datasets are easy to calculate according to the definition.
> * **ENKL**: The ENKL is designed to measure the relationship between two distributions, which corresponds to the two-class setting. To approximately estimate the ENKL on real datasets with multiple classes, we first group the node embeddings according to node labels, regarding them as observed samples of different class distributions. Further, we calculate the pair-wise ENKL among classes and report the weighted mean values as results.
> * **Settings**: For CMD, we report the fixed CM of the original data and the Weighted-CM of different operators. For ENKL, we report results of node embeddings after message passing (without feature transformation). We conduct the comparison of 4 operators (LP, HP, HTMP_0, HTMP_1) on 8 real-world datasets:
> $$
> \mathbf{H}^{LP}=\hat{\mathbf{A}}\mathbf{X},\quad
> \mathbf{H}^{HP}=(\mathbf{I}-\hat{\mathbf{A}})\mathbf{X},\\
> \mathbf{H}^{HTMP_0}=(\mathbf{I}-\hat{\mathbf{A}}+\hat{\mathbf{A}}^2)\mathbf{X},\quad
> \mathbf{H}^{HTMP_1}=(\mathbf{I}+\hat{\mathbf{A}}+\hat{\mathbf{A}}^2)\mathbf{X},\\
> $$
> The results are as follows:
> |Dataset|Ama.|Ama.|Rom.|Rom.|Cha.|Cha.|Squ.|Squ.|Act.|Act.|Fli.|Fli.|Blo.|Blo.|Pub.|Pub.|
> |-|-|-|-|-|-|-|-|-|-|-|-|-|-|-|-|-|
> |**MP Operator**|**CMD**|**ENKL**|**CMD**|**ENKL**|**CMD**|**ENKL**|**CMD**|**ENKL**|**CMD**|**ENKL**|**CMD**|**ENKL**|**CMD**|**ENKL**|**CMD**|**ENKL**|
> |**LP**|0.46|-0.26|0.85|-0.76|0.27|-25.38|0.08|-14.84|0.03|-2.77|0.40|-0.78|0.69|-0.41|1.40|-0.30|
> |**HP**|1.95|-0.80|2.77|-2.62|2.07|-126.50|2.02|-67.89|2.01|-9.28|1.88|-10.95|1.64|-16.01|0.71|-0.22|
> |**HTMP_0**|2.35|-1.14|3.31|-4.13|2.39|-150.75|2.22|-80.05|2.29|-11.53|1.93|-11.18|1.87|-16.48|2.00|-0.81|
> |**HTMP_1**|3.10|-2.41|3.55|-5.04|2.85|-170.64|2.28|-89.52|2.31|-11.63|2.67|-11.26|3.14|-16.77|4.73|-1.35|
>
> In line with expectations, the CMD and ENKL show an obvious negative correlation on all test datasets. This illustrates that **Theorem 3.2 still holds empirically in various real-world settings**.
>
>
>
> > **Weakness #2 and Question #2: The discussion of the connection and differences with existing analyses.**
>
> Answer #2:
> Thank you for this suggestion! The discussion of the Connections and differences with existing analyses is as follows:
>
> **Connections**:
> * CSBM is a widely used generative model to study the behavior of GNNs, with some variations [1-4]. To better study the graph hererophily, Luan et al. [3] introduced the homophily ratio into CSBM and proposed the CSBM-H and ENKL. **Our main theoretical analyses are based on the CSBM-H and ENKL.**
> * CM is proposed to describe the latent connection preference among classes within a graph [5,6]. It provides more information than the homophily ratio, thus is often utilized for heterophily study. Zhu et al. [7] highlighted that node degrees and CMs are key factors for message passing under heterophily. Similarly, Ma et al. [1] proposed the existence of a special case of heterophily where the homophily ratios remain low, but the VMP mechanism can achieve strong performance. **One of our conclusions in lines 127-129 is consistent with their findings.**
>
> **Differences**:
> * We newly introduce the **HTMP operator** in the CSBM-H to investigate the behavior of heterophilic GNNs, which is beyond existing analyses.
> * We newly propose the **CMD** to measure the discriminability of CM, providing a **quantitative metric** to measure the quality of CM.
> * Further, we theoretically prove the **negative correlation between CMD and ENKL** of VMP, providing **formal proof** for the findings of existing works.
> * The ways of existing works to utilize CM are limited in studying what kind of data is suitable for the message-passing mechanism, which is the **data** perspective. We take a further step with the **model** perspective that the CM can be enhanced by the HTMP mechanism, and then propose the **Weighted-CM** for better description.
> * Through empirical evaluation, we find that **increasing CMD is a possible reason for the effectiveness of HTMP**.
>
> In summary, our work is based on the existing CSBM-H, ENKL, and CM, and we further proposed the CMD, Weighted-CM, along with other new theoretical and empirical findings. We will update the above discussion in the revised version.
>
> Reference:
>
> [1] Is homophily a necessity for graph neural networks? ICLR 2022.
>
> [2] Understanding nonlinearity in graph neural networks from the bayesian-inference perspective. NeurIPS 2022.
>
> [3] When do graph neural networks help with node classification? investigating the homophily principle on node distinguishability. NeurIPS 2023.
>
> [4] Understanding Heterophily for Graph Neural Networks. ICML 2024.
>
> [5] Beyond homophily in graph neural networks: Current limitations and effective designs. NeurIPS 2020.
>
> [6] Graph neural networks with heterophily. AAAI 2021.
>
> [7] Heterophily and graph neural networks: Past, present and future. IEEE Data Eng. Bull. 2023.
>
>
>
> > **Question #1: The training/optimization dynamics of the compatibility matrix.**
>
> Answer #3: We show the training dynamics of the compatibility matrix and GNN predictions approximately as follows:
> * For CM, we use the corresponding **CMD** to better show its trend of change since the matrix is hard to visualize.
> * For GNN prediction, we report **the accuracy of the pseudo labels** (pse_acc).
> * Since the CM and GNN predictions can be updated many times during training, we choose some of them to show the dynamics.
>
> The results are as follows:
> |Dataset|Rom.|Rom.|Ama.|Ama.|Cha.|Cha.|Pub.|Pub.|
> |:-:|:-:|:-:|:-:|:-:|:-:|:-:|:-:|:-:|
> |**Update Step**|**CMD**|**pse_acc**|**CMD**|**pse_acc**|**CMD**|**pse_acc**|**CMD**|**pse_acc**|
> |Init.|0.740|0.501|0.456|0.574|0.424|0.564|1.391|0.587|
> |1|0.739|0.503|0.448|0.619|0.364|0.621|1.119|0.676|
> |2|0.434|0.554|0.242|0.670|0.341|0.584|1.113|0.693|
> |3|0.501|0.554|0.275|0.683|0.318|0.591|0.997|0.699|
> |4|0.575|0.647|0.286|0.695|0.187|0.594|0.847|0.711|
> |5|0.594|0.664|0.370|0.698|0.225|0.587|0.789|0.730|
> |10|0.608|0.714|0.424|0.716|0.246|0.659|0.847|0.808|
> |15|0.647|0.739|0.481|0.726|0.254|0.669|1.062|0.822|
> |20|0.687|0.755|0.535|0.731|0.418|0.674|1.168|0.839|
> |30|0.752|0.780|0.604|0.738|0.444|0.679|1.205|0.844|
> |40|0.806|0.798|0.622|0.741|-|-|1.211|0.847|
> |50|0.807|0.800|0.623|0.743|-|-|1.234|0.852|
> |100|0.826|0.858|-|-|-|-|1.383|0.935|
> |150|0.844|0.901|-|-|-|-|-|-|
> |Last.|0.859|0.919|0.626|0.746|0.488|0.682|1.507|0.938|
>
> As the training process progresses, **the accuracy of GNN prediction gradually increases, while the CMD first declines and then rises**.
> This is because the CMs are initialized mainly based on the training set labels at the beginning, since other predicted labels have extremely low confidence, which corresponds to low contributions.
> The inaccurate predictions may impact the CM estimation for a while at the early stage, leading to the reduction of CMD.
> Then, CMD will gradually increase along with the accuracy of GNN prediction, and enhance each other.
>
>
>
> > **Question #3: The detailed investigation focuses on low-degree and low label rate setting.**
>
> Answer #4: Thank you for the suggestion! The case studies on low-degree nodes and low label rate settings are indeed important to prove the effectiveness of CMGNN.
> * Thus, we have provided an investigation on the performance of nodes with various levels of degrees in **Appendix G.3**. The results show that **CMGNN can achieve outstanding performances on low-degree nodes**.
> * In addition, we conduct an experiment to investigate the performance of CMGNN under the low label rate setting. The results show that **CMGNN can still outperform strong baselines, demonstrating the effectiveness of CMGNN on handling the noise issue**.
>
> The details of the experiment are as follows:
> * Dataset: We conduct experiments on Roman-Empire, Amazon-Ratings, Chameleon-F, and Pubmed. **The label rate for training is set as 5% to meet the low label rate setting.**
> * Baseline: We compare CMGNN with 2 base and 3 top-performance baseline methods, including MLP, GCN, GCNII, ACMGCN, and OrderedGNN.
>
> The classification accuracies of methods are as follows:
>
> |Method|Roman-Empire|Amazon-Ratings|Chameleon-F|Pubmed|
> |-|-|-|-|-|
> |MLP|58.14±1.40|37.03±0.43|36.74±2.97|83.07±0.55|
> |GCN|33.53±0.77|37.32±0.52|36.91±2.22|83.95±0.62|
> |GCNII|63.27±0.72|39.93±0.87|38.40±3.27|**86.04±0.74**|
> |ACMGCN|59.93±2.03|39.96±0.81|37.43±2.84|85.54±0.79|
> |OrderedGNN|64.76±2.20|40.00±0.90|38.91±2.57|85.59±0.68|
> |CMGNN|**65.93±2.17**|**40.02±0.86**|**40.11±2.82**|85.62±0.94|
>
> Consistent with the results reported in the paper, CMGNN can also **achieve outstanding performance on the low label rate setting**.
>
>
>
> **We are expecting these could help answer your questions and look forward to further discussion. If you have any further questions, please feel free to contact us.**

---

> ### Comment · Reviewer_ZGc9 · 2025-08-02
>
> I thank the authors for providing an extensive followup on each of my raised points. In summary, the authors addressed (1) the practicality and feasibility of Theorem 3.2 as it pertains to real-world settings, (2) the relation between the analysis and existing analyses, (3) analyses and discussion of the joint training and optimization of the compatibility matrix and GNN, and (4) verification of the benefits of the approach in low label rate and low-degree settings. As such, I am happy to maintain my positive score.

---

> > ### Author Response · Authors · 2025-08-03
> >
> > Thank you for your positive feedback! We greatly appreciate your insightful and constructive comments, which have significantly improved the quality of our work. We will incorporate these changes in the final version of the manuscript.

---

### Official Review · Reviewer_SFcH · 2025-06-24

**Clarity:** 2
**Significance:** 3
**Originality:** 2
**Rating:** 4
**Confidence:** 4

**Summary:**

This paper investigates why message passing remains effective for Graph Neural Networks on heterophilic graphs, introducing the concept of Compatibility Matrix Discriminability (CMD) to explain the connection between message passing and node discriminability. Building on these insights, the authors propose CMGNN, a novel model that explicitly enhances CMD and demonstrates superior performance on heterophilic graphs across extensive benchmarks.

**Questions:**

1. What is the definition of HTMP method? Is it just the truncated Krylov method [1], i.e. identity + multi-hop neighborhood information? It seems that you are studying the normal message passing all the time.

2. "The high-pass operator is often used to capture the difference in heterophilic graphs." This is not accurate. High-pass operator computes the dissimilarity between the central node features and its aggregated neighborhood features, or conduct the neighborhood diversification operation [2].

3. What do you use $\dot{A}$ in equation (6)? Why not $\hat{A}$?

4. Why do we need CMD when we already have ENKL to measure node distinguishability?

5. Could you please demonstrate how sparsity will be harmful to MP theoretically (probably with toy example) or empirically with real-world datasets?

6. The writing for "Sufficient Neighborhood Supplement" is confusing. Need to re-organize the contents, clarify the notations and their dimensions.

7. More ablation studies on different proposed model components are needed are needed.





[1] Break the ceiling: Stronger multi-scale deep graph convolutional networks. Advances in neural information processing systems. 2019;32.


[2] Complete the missing half: Augmenting aggregation filtering with diversification for graph convolutional networks. arXiv preprint arXiv:2008.08844. 2020 Aug 20.

**Ethical Concerns:**

["NO or VERY MINOR ethics concerns only"]

**Final Justification:**

After going through the discussion from other reviewers, I'll keep my positive score.

**Limitations:**

yes

**Quality:**

3

**Strengths And Weaknesses:**

# Strengths

Quality: good \
Clarity: medium\
Significance: good\
Originality: medium

# Weaknesses

See below

---

> ### Author Rebuttal · Authors · 2025-07-31
>
> We would like to thank you for your comprehensive review. We have carefully considered your comments and suggestions, and the following are our detailed responses.
>
>
> > **Question #1: What is the definition of HTMP method? Is it just the truncated Krylov method?**
>
> Answer #1: Thank you for the reminder. **We use "HTMP" methods to refer to the heterophilic GNNs that follow the message-passing mechanism.** As mentioned in Section 2, HTMP methods often extend the neighborhood variously and redesign the COMBINE function to collect neighbor messages preferentially while preserving the ego messages.
>
> Thus, the truncated Krylov method can be seen as **a special case of HTMP methods** since it has a similar way to conduct message passing. Further, the HTMP methods are not limited to the truncated Krylov method. The definition of extended neighborhood is **more than the multi-hop neighborhood**. It can also be extended to some flexible neighbor types, such as feature-similarity-based neighborhoods [1] and global neighborhoods [2].
>
> For theoretical analysis, we start from the vanilla message passing and then **extend it to heterophilic message passing**. Since it can have various forms, we choose a typical scheme as the example, and it indeed can be seen as a special case of the truncated Krylov method. In addition, we study the representative HTMP methods, i.e., GloGNN, GPR-GNN, and ACM-GCN empirically.
>
> Reference:
>
> [1] Node similarity preserving graph convolutional networks. WSDM 2021.
>
> [2] Finding global homophily in graph neural networks when meeting heterophily. ICML 2022.
>
>
>
> > **Question #2: A more accurate description of the high-pass operator.**
>
> Answer #2: Thank you for the correction! Our intention is also to claim that the high-pass operator can capture the difference between the central node and its neighborhoods, but with a brief description, which may not be clear enough. We will improve it and update the citation in the revised version.
>
>
>
> > **Question #3: Why do you use $\dot{\mathbf{A}}$ in equation (6)? Why not $\hat{\mathbf{A}}$?**
>
> Answer #3: $\dot{\mathbf{A}}$ is an indicator matrix of the practical aggregate weights matrix during message passing. It can be **various** in different message passing methods. For example, $\dot{\mathbf{A}}=\hat{\mathbf{A}}$ in low-pass filtering and $\dot{\mathbf{A}}=\mathbf{I}-\hat{\mathbf{A}}$ for high-pass filtering. In many heterophilic message passing methods, neighbors may have **different contributions** to the embedding of the central node. Thus, **the aggregated messages meet a practical Weighted-CM different from the fixed CM**. To better describe the practical CM of different message passing methods, we introduce the $\dot{\mathbf{A}}$ and the Weighted-CM.
>
>
>
> > Question #4: Why do we need CMD when we already have ENKL to measure node distinguishability?
>
> Answer #4: First, ENKL measures the relationship between two distributions, which is designed for the **two-class setting.** Second, it is difficult to estimate the node embedding distribution of real-world datasets. In comparison, CMD focuses on the coupling of node labels and graph structure. It fits the multi-class settings and is easy to calculate.
>
>
>
> > **Question #5: How sparsity will be harmful to MP theoretically (probably with a toy example) or empirically with real-world datasets?**
>
> Answer #5: **Sparsity may cause an outlier neighborhood for nodes, leading to embedding overlap among classes after message passing.** We give a toy example to illustrate this process.
>
> Considering a graph with nodes belonging to 3 classes. The latent connection preference among classes follows the compatibility matrix:
> $$
> \mathbf{M}=\\begin{bmatrix}
> 0.3,&0.1,&0.6\\\\
> 0.1,&0.7,&0.2\\\\
> 0.6,&0.2,&0.2
> \\end{bmatrix}.
> $$
> For simplicity, the features of nodes in the same class are set to the same:
> $$
> \mathbf{X}\_{c0}=(1, 0, 0),\\quad
> \mathbf{X}\_{c1}=(0, 1, 0),\\quad
> \mathbf{X}\_{c2}=(0, 0, 1).
> $$
>
> We use a vanilla message passing $\mathbf{H}=\hat{\mathbf{A}}\mathbf{X}$ as an example.
> Suppose we have two nodes $u$ and $v$, belonging to class 0 and class 1, respectively.
>
> If node degrees are large enough, e.g., 10. Then node $u$ may have 3 neighbors from class 0, 1 from class 1 and 6 from class 2. Similarly, node $v$ may have 1 neighbor from class 0, 7 from class 1 and 2 from class 2. Thus, the node embeddings are as follows after message passing:
> $$
> \mathbf{h}\_{u}=(0.3, 0.1, 0.6),\quad
> \mathbf{h}\_{v}=(0.1, 0.7, 0.2),
> $$
> which can be easily distinguished.
>
> Further, we consider the sparsity issue. Now, nodes $u$ and $v$ can only have 2 neighbors each due to sparsity. Therefore, there might be a situation where nodes $u$ and $v$ have similar neighborhoods. For example, node $u$ has 1 neighbor from class 0 and 1 from class 2, while node $v$ also has 1 neighbor from class 0 and 1 from class 2. Thus, the node embeddings after message passing are:
> $$
> \mathbf{h}\_{u}=(0.5, 0, 0.5),\quad
> \mathbf{h}\_{v}=(0.5, 0, 0.5).
> $$
> Thus, **two nodes from different classes may share similar embeddings after message passing in sparsity settings** and it's difficult to distinguish them.
>
> In addition, we have conducted experiments on synthetic datasets to investigate the influence of node degrees in Appendix C. The results also show that low degrees have a negative impact on the performance of GCN, which is consistent with the above analysis.
>
>
>
> > **Question #6: The writing for "Sufficient Neighborhood Supplement" is confusing. Need to re-organize the contents, clarify the notations and their dimensions.**
>
> Answer #6: Thanks for your suggestion! This is indeed that the "Sufficient Neighborhood Supplement" may not be clear enough. In this component, we additionally introduce $K$ virtual neighbors for each node to alleviate the sparsity issue. Thus, "Virtual Neighborhood Supplement" might be relatively clearer. We will further improve this part in the revised version.
>
>
>
> > **Question #7: More ablation studies on different proposed model components are needed.**
>
> Answer #7: We further conduct ablation studies on several components on all datasets, including Confidence-based CM Estimation (CE), degree weighting function (DW) and adaptive weighting message combine (AW), where the latter two are introduced in Appendix E.
> * "W/O CE" means that we remove the confidence during the CM estimation.
> * "W/O DW" means that we remove the degree weighting function during the CM estimation.
> * "W/O AW" means that we replace the adaptive weighting with the mean operator in the message combination.
>
> The complete results are as follows:
>
> |Variants|Rom.|Sna.|Squ.|Act.|Fli.|Cha.|Ama.|Blog|Pok.|Pen.|Twi.|Gen.|Pub.|
> |-|-|-|-|-|-|-|-|-|-|-|-|-|-|
> |**CMGNN**|84.35±1.27|59.86±0.61|41.89±2.34|36.82±0.78|92.66±0.46|45.70±4.92|52.13±0.55|97.00±0.52|81.42±0.55|83.01±0.48|65.18±0.31|85.19±0.53|89.99±0.32|
> |W/O CE|83.88±1.41|59.54±0.77|40.35±2.43|36.47±1.22|92.51±0.51|44.75±3.05|51.93±0.38|96.72±0.51|80.67±0.65|82.58±0.49|65.09±0.26|85.03±0.47|89.68±0.36|
> |W/O SM|83.82±1.29|59.48±0.78|40.72±2.28|36.05±1.24|92.22±0.47|42.29±4.38|51.91±0.83|96.61±0.44|79.39±0.37|81.68±1.55|65.01±0.34|84.94±0.56|89.53±0.41|
> |W/O DL|83.63±1.35|59.68±0.72|41.65±2.55|36.41±1.08|92.66±0.46|44.92±4.12|52.05±0.57|97.00±0.52|80.28±0.49|81.34±1.83|64.88±0.20|84.87±0.75|89.99±0.32|
> |W/O CE and DL|83.77±1.38|59.46±0.71|39.80±2.36|36.32±1.05|92.28±0.52|44.58±3.28|51.74±0.55|96.53±0.53|79.26±0.34|81.20±0.61|64.64±0.19|84.53±0.59|88.97±0.48|
> |W/O SM and DL|83.48±1.89|59.25±0.75|40.19±2.69|35.66±1.42|91.98±0.49|41.01±3.09|51.49±1.02|96.39±0.58|77.57±0.59|80.71±0.48|64.33±0.32|84.28±0.50|88.71±0.45|
> |W/O DW|83.80±1.68|59.62±0.72|41.61±2.39|36.45±1.26|92.51±0.56|44.25±5.06|52.08±0.52|96.85±0.51|80.82±0.72|82.69±0.57|65.04±0.38|85.13±0.56|89.93±0.32|
> |W/O AW|78.88±4.03|58.12±0.65|36.40±2.95|36.56±1.21|90.77±0.44|42.01±3.70|51.21±0.74|96.23±0.51|80.13±0.34|81.66±0.42|64.84±0.22|84.39±0.15|89.73±0.42|
>
>
> The results demonstrate the effectiveness of these components.
> * For SM and DL, the results are **consistent with the results reported in the paper**, except that DL shows no significant improvement on some datasets. This may be due to the discriminability of SM reaching the bottlenecks, and can not be further improved by DL.
> * For CE, we observe that "W/O CE" shows performance drops on all datasets, which indicates that **CE is effective in alleviating the issue of noise**.
> * For DW and AW, we can find that they **both have positive contributions to the performance**. DW has a relatively low impact, while AW makes significant performance differences on some datasets. This may be due to the high demand for ego messages on heterophilic graphs.
>
>
>
> **We are expecting these could help answer your questions and look forward to further discussion. If you have any further questions, please feel free to contact us.**

---

### Official Review · Reviewer_eeXf · 2025-06-29

**Clarity:** 2
**Significance:** 2
**Originality:** 2
**Rating:** 3
**Confidence:** 4

**Summary:**

This paper explores the connection between the discriminability of node embedding and the compatibility matrix (CM), and attributes the effectiveness of heterophilic message passing to increasing the proposed Compatibility Matrix Discriminability (CMD). Building on this insight, the authors introduce CMGNN, a novel framework that explicitly improves CMD to enhance node embeddings, while alleviating the issues of graph sparsity and noise.

**Questions:**

1. Why GraphSAGE is not included in the comparison, given that it has demonstrated strong performance on both homophilic and heterophilic graphs in prior studies and our own experience?
2. Appendix E details engineering practices like a weighting function for CM estimation, additional structural features and adaptive weighting of three sources of messages. Can you provide ablation studies to quantify their individual contributions to CMGNN’s performance?
3. In Figure 6, the CMD value of Rom-Glo is exceptionally high, yet its performance reported in Table 1 is relatively poor. This discrepancy suggests that the relationship between CMD and model performance may be more nuanced than the paper implies, especially on datasets characterized by multiple classes, high heterophily, and low average degree. It would be helpful to clarify how CMGNN addresses such cases and whether CMD remains a reliable indicator of representation quality under these challenging conditions.
4. In Figure 7, the improvements in CMD achieved by the Weighted-CMs appear marginal compared to the original versions. This raises the question of how much the weighted design actually contributes to the overall performance gains. A quantitative analysis or ablation would help clarify the effectiveness of this component.

**Ethical Concerns:**

["NO or VERY MINOR ethics concerns only"]

**Limitations:**

yes

**Quality:**

3

**Strengths And Weaknesses:**

1. Extensive experiments have been conducted on multiple real-world and synthetic datasets with various homophily levels, node degrees and CMD levels, demonstrating the significance of CMD for heterophilic message passing and the robustness and cost-effectiveness of the proposed CMGNN.
2. A total of 19 representative homophilic and heterophilic GNN models have been integrated into a unified codebase and evaluated on 13 benchmark datasets with diverse proporties. The implementation details are also provided, facilitating reproducibility and further research in this area.
3. The presentation of experimental results lacks clarity in distinguishing performance across different homophily levels. Grouping the results (or at least the average ranks) by homophilic and heterophilic graphs would better highlight the effectiveness of the proposed CMGNN in each setting. Such stratified evaluation is particularly important for assessing heterophily-specific models.
4. While Theorem 3.2 provides insight into the negative correlation between CMD and ENKL, its assumptions—two-class setting, Gaussian feature distributions, ignored node degree effects, and reliance on the CSBM-H model—limit its direct applicability to real-world complex datasets. Although CMGNN mitigates some of these issues empirically, the paper lacks direct validation of the CMD-ENKL relationship on real datasets.
5. In Table 3, the ablation study on the effects of supplementary messages (SM) and discrimination loss (DL) is conducted on only five datasets. It remains unclear how these components perform on the remaining datasets. A more comprehensive evaluation across all datasets would provide a clearer understanding of their generalizability and overall contribution.

---

> ### Author Rebuttal · Authors · 2025-07-31
>
> We would like to thank you for your comprehensive review. We have carefully considered your comments and suggestions, and the following are our detailed responses.
>
> > **Weakness #1: The analysis of experimental results considering different homophily levels.**
>
> Answer #1: Thank you for pointing that out! We divide the graphs into two groups according to their edge homophily levels with a threshold of 0.5. The rank of CMGNN on various datasets and groups is as follows:
> |Datasets|Rom.|Sna.|Squ.|Act.|Fli.|Cha.|Ama.|Blog|Pok.|Pen.|Avg. Rank|
> |-|-|-|-|-|-|-|-|-|-|-|-|
> |Homo.|0.05|0.07|0.22|0.22|0.24|0.25|0.38|0.4|0.45|0.47|-|
> |CMGNN|1|1|1|4|1|1|1|1|1|1|1.3|
>
> |Datasets|Twi.|Gen.|Pub.|Avg. Rank|
> |-|-|-|-|-|
> |Homo.|0.55|0.62|0.8|-|
> |CMGNN|3|2|2|2.3|
>
> Further, the average ranks of baseline methods on heterophilic and homophilic groups are as follows:
>
> |Method|Hetero. Group|Homo. Group|Overall|
> |-|-|-|-|
> |MLP|15|14.7|14.9|
> |GCN|13.9|12.7|13.6|
> |GAT|13.1|12.7|13|
> |GraphSAGE|8.1|10|8.5|
> |APPNP|11.7|12.3|11.8|
> |GCNII|4.8|2.7|4.3|
> |H2GCN|14.7|14.7|14.7|
> |MixHop|7.1|8.3|7.4|
> |GBK-GNN|17.1|17|17.1|
> |GGCN|16.6|15.7|16.4|
> |GloGNN|9.9|4|8.5|
> |HOGGCN|13.8|19|15|
> |GPR-GNN|8|11.3|8.8|
> |ACM-GCN|6.6|11|7.6|
> |OrderedGNN|6.8|3.3|6|
> |M2MGNN|10|9|9.8|
> |N$^2$|9|7.3|8.6|
> |CLP|12.3|11|12|
> |EPFGNN|14.5|14|14.4|
> |CPGNN|13.3|13.7|13.4|
> |**CMGNN**|**1.3**|**2.3**|**1.5**|
>
> * From the results, we can find that **CMGNN shows significant effectiveness on heterophilic graphs** with an average rank of 1.3 and achieves the best on 9 of 10.
> * Also, **CMGNN can keep competitive performance on homophilic graphs** with an average rank of 2.3, which is also the best compared with baseline methods.
> * Interestingly, some **heterophilic GNNs** work relatively better on homophilic graphs rather than heterophilic graphs, such as GloGNN and OrderedGNN. This might be because these methods are relatively more inclined to adapt to both situations.
>
>
> > **Weakness #2: The lack of direct validation of the CMD-ENKL relationship on real datasets.**
>
> Answer #2: Thank you for this suggestion! A validation of the CMD-ENKL relationship on real datasets is indeed necessary. We have investigated it, and **the results show that the negative correlation between CMD and ENKL still holds in real-world datasets**.
>
> The details are as follows:
> * **CMD**: The CMDs of fixed CM and Weighted-CM on real datasets are easy to calculate according to the definition.
> * **ENKL**: The ENKL is designed for two-class setting. To approximately estimate the ENKL with multiple classes, we first group the node embeddings according to node labels and then calculate the pair-wise ENKL among classes and report the weighted mean values.
> * **Settings**: For CMD, we report the fixed CM and the Weighted-CM of different operators. For ENKL, we report results of node embeddings after message passing (without feature transformation). We conduct the comparison of 4 operators (LP, HP, HTMP_0, HTMP_1):
> $$
> H^{LP}=\hat{A}X,\quad
> H^{HP}=(I-\hat{A})X,\quad
> H^{HTMP_0}=(I-\hat{A}+\hat{A}^2)X,\quad
> H^{HTMP_1}=(I+\hat{A}+\hat{A}^2)X.
> $$
> The results are as follows:
> |Dataset|Ama.|Ama.|Rom.|Rom.|Cha.|Cha.|Squ.|Squ.|Act.|Act.|Fli.|Fli.|Blo.|Blo.|Pub.|Pub.|
> |-|-|-|-|-|-|-|-|-|-|-|-|-|-|-|-|-|
> |**Operator**|**CMD**|**ENKL**|**CMD**|**ENKL**|**CMD**|**ENKL**|**CMD**|**ENKL**|**CMD**|**ENKL**|**CMD**|**ENKL**|**CMD**|**ENKL**|**CMD**|**ENKL**|
> |**LP**|0.46|-0.26|0.85|-0.76|0.27|-25.38|0.08|-14.84|0.03|-2.77|0.40|-0.78|0.69|-0.41|1.40|-0.30|
> |**HP**|1.95|-0.80|2.77|-2.62|2.07|-126.50|2.02|-67.89|2.01|-9.28|1.88|-10.95|1.64|-16.01|0.71|-0.22|
> |**HTMP_0**|2.35|-1.14|3.31|-4.13|2.39|-150.75|2.22|-80.05|2.29|-11.53|1.93|-11.18|1.87|-16.48|2.00|-0.81|
> |**HTMP_1**|3.10|-2.41|3.55|-5.04|2.85|-170.64|2.28|-89.52|2.31|-11.63|2.67|-11.26|3.14|-16.77|4.73|-1.35|
>
> In line with expectations, **the CMD and ENKL show an obvious negative correlation on all test datasets**. This illustrates that Theorem 3.2 still holds empirically on real-world datasets.
>
>
> > **Weakness #3 & Question #2: A more comprehensive evaluation of ablation studies across all datasets including the implementation details.**
>
> Answer #3:
> Thank you for the suggestion! We investigate the influence of Confidence-based CM Estimation (CE), degree weighting function (DW), and adaptive weighting message combination (AW).
> * "W/O CE" means removing the confidence during the CM estimation.
> * "W/O DW" means removing the degree weighting function during the CM estimation.
> * "W/O AW" means replacing the adaptive weighting with the mean operator in the message combination.
> * The ablation study of additional structural features is provided in Appendix G.2.
>
> The complete results are as follows:
> |Variants|Rom.|Sna.|Squ.|Act.|Fli.|Cha.|Ama.|Blog|Pok.|Pen.|Twi.|Gen.|Pub.|
> |-|-|-|-|-|-|-|-|-|-|-|-|-|-|
> |CMGNN|84.35±1.27|59.86±0.61|41.89±2.34|36.82±0.78|92.66±0.46|45.70±4.92|52.13±0.55|97.00±0.52|81.42±0.55|83.01±0.48|65.18±0.31|85.19±0.53|89.99±0.32|
> |W/O CE|83.88±1.41|59.54±0.77|40.35±2.43|36.47±1.22|92.51±0.51|44.75±3.05|51.93±0.38|96.72±0.51|80.67±0.65|82.58±0.49|65.09±0.26|85.03±0.47|89.68±0.36|
> |W/O SM|83.82±1.29|59.48±0.78|40.72±2.28|36.05±1.24|92.22±0.47|42.29±4.38|51.91±0.83|96.61±0.44|79.39±0.37|81.68±1.55|65.01±0.34|84.94±0.56|89.53±0.41|
> |W/O DL|83.63±1.35|59.68±0.72|41.65±2.55|36.41±1.08|92.66±0.46|44.92±4.12|52.05±0.57|97.00±0.52|80.28±0.49|81.34±1.83|64.88±0.20|84.87±0.75|89.99±0.32|
> |W/O CE and DL|83.77±1.38|59.46±0.71|39.80±2.36|36.32±1.05|92.28±0.52|44.58±3.28|51.74±0.55|96.53±0.53|79.26±0.34|81.20±0.61|64.64±0.19|84.53±0.59|88.97±0.48|
> |W/O SM and DL|83.48±1.89|59.25±0.75|40.19±2.69|35.66±1.42|91.98±0.49|41.01±3.09|51.49±1.02|96.39±0.58|77.57±0.59|80.71±0.48|64.33±0.32|84.28±0.50|88.71±0.45|
> |W/O DW|83.80±1.68|59.62±0.72|41.61±2.39|36.45±1.26|92.51±0.56|44.25±5.06|52.08±0.52|96.85±0.51|80.82±0.72|82.69±0.57|65.04±0.38|85.13±0.56|89.93±0.32|
> |W/O AW|78.88±4.03|58.12±0.65|36.40±2.95|36.56±1.21|90.77±0.44|42.01±3.70|51.21±0.74|96.23±0.51|80.13±0.34|81.66±0.42|64.84±0.22|84.39±0.15|89.73±0.42|
>
> The results demonstrate the effectiveness of these components.
> * For SM and DL, the results are **consistent with the results reported in the paper**, except that DL shows no significant improvement on some datasets. This may be due to the discriminability of SM reaching the bottlenecks, and can not be further improved by DL.
> * For CE, we observe that "W/O CE" shows performance drops on all datasets, which indicates that **CE is effective in alleviating the issue of noise**.
> * DW and AW **both have positive contributions to the performance**. DW has a relatively low impact, while AW makes significant performance differences on some datasets. This may be due to the high demand for ego messages on heterophilic graphs.
>
>
> > **Question #1: Why GraphSAGE is not included in the comparison?**
>
> Answer #4:
> Thank you for the supplement! We simply chose some of the representative homophilic GNNs as baseline methods rather than all of them since this paper focuses more on the heterophilic setting.
> We have added GraphSAGE for comparison and its performance is as follows:
> |Datasets|Rom.|Sna.|Squ.|Act.|Fli.|Cha.|Ama.|Blog|Pok.|Pen.|Twi.|Gen.|Pub.|Hetero. Rank|Homo. Rank|Overall Rank|
> |-|-|-|-|-|-|-|-|-|-|-|-|-|-|-|-|-|
> |GraphSAGE|69.62±1.40|35.72±0.12|38.13±1.71|36.12±1.40|92.00±0.58|42.18±4.64|45.07±0.54|96.30±0.44|77.22±0.06|78.90±0.37|62.14±0.09|84.37±0.15|88.86±0.56|8.1|10|8.5|
>
> **Indeedly, GraphSAGE performs well in both homophilic and heterophilic graphs.** It significantly outperforms GCN and GAT and even many heterophilic GNNs. It further enhances our evaluation, and we will update the corresponding part in the revised version.
>
>
> > **Question #3: The relationship between CMD and model performance on real-world datasets. How CMGNN addresses such cases and whether CMD remains a reliable indicator.**
>
> Answer #5: CMD measures the discriminability considering the graph structure and labels, without node features.
> The quality of node embeddings after message passing relies on both CMD and node features.
> **CMD can play a reliable indicator in real-world datasets when the node features are fixed as shown in Answer #2.** However, the feature transformations are different in the methods. Thus, the high CMD and relatively poor performance of Rom-Glo may be due to **its feature transformation being unsatisfactory**. Even extremely high CMD can not ensure the good performance of Rom-Glo.
>
> To address real-world issues, CMGNN constructs supplementary neighbors for each node and utilizes the discrimination loss to directly enhance the discriminability of node embeddings.
>
>
> > **Question #4: How much does the Weighted-CM contribute to the overall performance gains?**
>
> Answer #6: To show the detailed contribution of Weighted-CM, we conduct an ablation study. For "W/O WCM", we use the fixed CM of the original data to replace the Weighted-CM during CM-aware message passing. The results are as follows:
> |Variants|Rom.|Sna.|Squ.|Act.|Fli.|Cha.|Ama.|Blog|Pok.|Pen.|Twi.|Gen.|Pub.|
> |-|-|-|-|-|-|-|-|-|-|-|-|-|-|
> |**CMGNN**|84.35±1.27|59.86±0.61|41.89±2.34|36.82±0.78|92.66±0.46|45.70±4.92|52.13±0.55|97.00±0.52|81.42±0.55|83.01±0.48|65.18±0.31|85.19±0.53|89.99±0.32|
> |W/O WCM|83.55±2.40|59.38±0.47|41.12±1.96|36.19±1.05|92.40±0.50|44.75±4.25|51.88±0.63|96.67±0.51|81.20±0.62|82.79±0.52|65.01±0.49|84.89±0.62|89.82±0.36|
>
> The results show that the **Weighted-CMs can achieve better performance compared with the "W/O WCM" setting, which uses the exact CM of the original data rather than estimating**.
> The performance difference is relatively insignificant on some datasets. This might be because the other components of CMGNN, such as SM and CE, have already made significant contributions to learn better node embeddings.
>
>
> **We are expecting these could help answer your questions and look forward to further discussion. If you have any further questions, please feel free to contact us.**

---

> > ### Comment · Reviewer_eeXf · 2025-08-07
> >
> > Thank you for your detailed responses to the review comments. The additional experiments and analyses you provided have addressed several of the initial concerns. There are some remaining concerns based on the rebuttal:
> >
> > 1. **Nuanced CMD-Performance Relationship:** While the rebuttal attributes Rom-Glo's poor performance despite high CMD to feature transformation issues, this reveals CMD's insufficiency as a standalone performance predictor.
> > 2. **Weighted-CM’s Marginal Gains:** The ablation shows Weighted-CM’s improvements are modest on some datasets. The rebuttal attributes this to other components’ dominance, but a deeper analysis could better contextualize its value.
> > 3. **Theoretical Assumptions vs. Real-World Complexity:** While the empirical results in the rebuttal support Theorem 3.2's negative correlation between CMD and ENKL, the theorem itself relies on strong assumptions. The current validation remains post hoc—it demonstrates alignment with the theory but does not address why CMGNN works under more complex conditions. Without a deeper theoretical discussion, the paper risks presenting CMD as a heuristic rather than a principled framework. We encourage the authors to either generalize the theory or explicitly discuss its limitations and CMGNN’s empirical adaptations to bridge the gap.

---

> > > ### Author Response · Authors · 2025-08-08
> > > **Reply to the Comment of Reviewer eeXf**
> > >
> > > We sincerely thank you for the thoughtful and constructive comments. We are pleased to further discuss your remaining concerns.
> > >
> > > > **Concern #1: Nuanced CMD-Performance Relationship.**
> > >
> > > **Response #1:**
> > > **CMD primarily examines the relationship between graph structure and node labels**, without considering the effect of node features. Therefore, it may not serve as a standalone performance indicator naturally. Instead, it remains a valuable metric when feature transformations are fixed.
> > >
> > > ---
> > > > **Concern #2: Weighted-CM’s Marginal Gains.**
> > >
> > > **Response #2:**
> > > Thank you for this suggestion! We conducted additional ablation studies to evaluate the contributions of SM and DL in the W/O WCM setting. The results are summarized below:
> > > |Variants|Rom.|Sna.|Squ.|Act.|Fli.|Cha.|Ama.|Blog|Pok.|Pen.|Twi.|Gen.|Pub.|
> > > |-|-|-|-|-|-|-|-|-|-|-|-|-|-|
> > > |**CMGNN**|84.35±1.27|59.86±0.61|41.89±2.34|36.82±0.78|92.66±0.46|45.70±4.92|52.13±0.55|97.00±0.52|81.42±0.55|83.01±0.48|65.18±0.31|85.19±0.53|89.99±0.32|
> > > |W/O WCM|83.55±2.40|59.38±0.47|41.12±1.96|36.19±1.05|92.40±0.50|44.75±4.25|51.88±0.63|96.67±0.51|81.20±0.62|82.79±0.52|65.01±0.49|84.89±0.62|89.82±0.36|
> > > |W/O WCM and SM|83.12±1.98|58.97±0.92|40.29±2.41|36.08±0.82|91.86±0.63|42.15±3.94|51.33±0.65|96.59±0.49|79.47±0.63|81.57±1.21|64.78±0.57|84.49±0.48|89.41±0.41|
> > > |W/O WCM and DL|83.06±1.85|59.08±0.67|39.73±2.47|36.12±1.14|92.08±0.48|43.85±4.23|51.68±0.69|96.61±0.54|80.13±0.67|81.30±1.61|64.75±0.21|84.44±0.63|89.62±0.38|
> > >
> > > These results demonstrate that both SM and DL provide meaningful contributions in the W/O WCM setting. Notably, when Weighted-CM gains are marginal (e.g., on Pokec and Penn94), **further removal of SM and DL leads to significant performance degradation**. This aligns with our design philosophy: **SM and DL first address the gap** between theoretical assumptions and real-world challenges (imbalanced degrees and low-quality features), while **Weighted-CM serves as an enhancement** to further optimize the possible low-quality CM, like the icing on the cake.
> > >
> > > ---
> > > > **Concern #3: Theoretical Assumptions vs. Real-World Complexity.**
> > >
> > > **Response #3:**
> > > Although Theorem 3.2 relies on strong assumptions, we have empirically demonstrated its applicability in real-world graphs, which significantly exceed these assumptions and are difficult to describe theoretically.
> > > As suggested, we have explicitly discussed the gaps between assumptions and real-world graphs, along with the corresponding empirical adaptations of CMGNN:
> > > * **Mutil-class Setting**: CM and CMD naturally meet the multi-class setting, offering an advantage over the homophily ratio, which can only describe the connection between the same class or not. Thus, **no additional design is needed** for CMGNN in the multi-class setting.
> > > * **Semi-Supervised Setting**: Node labels are not all available in the semi-supervised setting. Thus, CMGNN introduces the **confidence-based CM estimation** to address the noise issue.
> > > * **Low/Imbalanced Node Degrees**: In real-world graphs, node degrees can be low and imbalanced. First, **CMGNN introduces the supplementary neighbors and adaptively learns the contribution of supplementary messages** for each node. Second, CMGNN has a **degree-weighted function** during CM estimation, considering the imbalanced node degrees.
> > > * **Potential Low-Quality Node Features**: The node features do not follow the Gaussian distribution and may be low-quality in real-world graphs. Thus, CMGNN proposes **a discrimination loss to directly increase the discriminability of node embeddings**.
> > >
> > > ---
> > >
> > > We hope the above responses have adequately addressed your concerns. If you find our work valuable and these responses have alleviated your concerns, we would be most grateful if you could kindly reconsider the final rating. Should any further questions arise, we remain very willing to clarify any points during the discussion process.

---

### Official Review · Reviewer_tib1 · 2025-06-29

**Clarity:** 3
**Significance:** 2
**Originality:** 2
**Rating:** 4
**Confidence:** 4

**Summary:**

This paper examines the effectiveness of message-passing mechanisms in heterophilous graph neural networks (HTGNNs), where connected nodes often display contrasting behaviors. This paper propose a unified heterophilous message-passing (HTMP) mechanism and introduce CMGNN, which enhances the compatibility matrix to address incomplete and noisy semantic neighborhoods.

**Questions:**

See the Weakness.

**Ethical Concerns:**

["NO or VERY MINOR ethics concerns only"]

**Final Justification:**

My main concern, the doubts regarding the motivation of the paper have been resolved, therefore I have decided to increase the score.

**Limitations:**

Yes

**Quality:**

3

**Strengths And Weaknesses:**

I have previously reviewed this paper. Compared to the earlier version I evaluated, the authors have included experiments on large datasets to demonstrate the scalability of their method. However, some issues still remain unresolved.

**Strengths:**

1.  The superior performance of CMGNN is substantiated through comprehensive evaluations across 13 varied datasets, benchmarked against 19 existing methods.

**Weaknesses:**

1.  **Motivation**: The introduced HTMP framework does not seem to be specifically designed for the problem of heterophily. Its core components, such as FUSE and COMBINE, are common techniques in the GNN field (e.g., used to address over-smoothing or over-squashing) rather than being exclusive solutions for heterophily, making the framework appear more like a generalized enhancement of the message-passing paradigm.

2.  **Novelty**: The concept of a Compatibility Matrix has been previously explored in GNNs. This work does not seem to offer any new methodological contributions to it, which casts doubt on its originality.

3.  **Methodology**

    - The paper fails to provide a clear rationale for the specific architectural choices within AGG, COMBINE, and FUSE, leaving it unclear how these designs contribute to a more effective estimation of the compatibility matrix by the HTMP framework.

    - Furthermore, the construction of these components seems excessively handcrafted and arbitrary. For instance, it is not justified why the COMBINE step integrates information from only the three specified neighbor types, or why FUSE employs concatenation while COMBINE uses a weighted sum, giving the impression of an ad-hoc assembly of pre-existing techniques.

---

> ### Author Rebuttal · Authors · 2025-07-31
>
> We sincerely appreciate your valuable comments and suggestions. Although you mentioned that you have already compared the new version, we would like to kindly remind you that **the weaknesses you pointed out have been significantly and explicitly addressed in the current revision**. Specifically,
>
> * For Weakness #1, **the HTMP framework has been totally removed in this version**, since it has relatively little connection with the theme of this paper.
> * For Weakness #2, in this version, we have **newly proposed the Compatibility Matrix Discriminability (CMD)** to measure the discriminability of CM. Further, we theoretically and empirically prove the positive correlation between node discriminability and CMD.  In addition, a toy example and a posterior evaluation in representative HTGNNs show that the effectiveness of existing heterophilic message-passing mechanisms may be attributed to increasing CMD, which leads to better node embeddings.
> * For Weakness #3, the motivation behind method design is to address the issues of sparsity and noise in real-world graphs, with a detailed description in lines 168-177 and Section 4.
>
> Based on the above improvement, we believe **the weakness of the previous submission does not match this version**, especially the parts that have been totally removed. We appreciate your time for reviewing our paper and providing feedback, but we also hope the improvement can be considered fairly and justly.
>
> If you have any further questions, please feel free to contact us.

---

> > ### Comment · Reviewer_tib1 · 2025-08-05
> >
> > Thank you to the author for the reply and the effort made. Considering that the additional experiments and efforts during the rebuttal (including those for other reviewers) have made the paper more solid, I have decided to increase my score.

---

> > > ### Author Response · Authors · 2025-08-05
> > >
> > > Thank you for your positive feedback and for increasing your score! Once again, we sincerely appreciate your thoughtful and constructive comments, which played a crucial role in improving this paper.

---

### Official Review · Reviewer_5t7f · 2025-07-01

**Clarity:** 3
**Significance:** 3
**Originality:** 3
**Rating:** 5
**Confidence:** 4

**Summary:**

This paper investigates why message passing mechanisms in Graph Neural Networks (GNNs) remain effective on heterophilic graphs, where connected nodes tend to belong to different classes. The authors introduce the concept of Compatibility Matrix Discriminability (CMD), proposing that the success of message passing in such settings can be attributed to an increase in CMD which means better discriminability in CM. Building on this insight, they propose CMGNN, a new method that enhances CMD through a combination of sufficient neighborhood supplementation, confidence-based compatibility matrix estimation, and CM-aware message passing. The model is evaluated on 13 datasets against 19 baselines, showing superior performance.

**Questions:**

1、Since the model needs to do CM estimation under semi-supervised settings, how sensitive is CMGNN to the choice or accuracy of pseudo labels used in compatibility matrix estimation?
2、In evolving graphs (new nodes or labels arrive over time), would re-estimating CMD and retraining CMGNN be feasible, or is the method inherently static?
3、Many real-world semi-supervised graphs have imbalanced class distributions. Does CMD offer a reliable signal under label imbalance?

**Ethical Concerns:**

["NO or VERY MINOR ethics concerns only"]

**Final Justification:**

Most of my main concerns have been well addressed and the authors' effort during the rebuttal phase has made this work much more solid, I will increase my **confidence score from 3 to 4**.

**Limitations:**

Yes

**Paper Formatting Concerns:**

No concern

**Quality:**

3

**Strengths And Weaknesses:**

The paper is well-motivated and logically structured, guiding the reader from motivation to methodology and results. The authors provide a theoretical analysis connecting Compatibility Matrix Discriminability (CMD) with node discriminability (ENKL), showing that the increase in CMD is a possible reason for the effectiveness of heterophilic message-passing. And extensive experiments of the proposed CM-aware GNN against baselines further validate their findings.

However, the theoretical results rely heavily on the CSBM-H model, which may not fully represent real-world graph structures. Although the method performs well, it introduces extra computation, which might become costly on dynamic or real-time graphs.

---

> ### Author Rebuttal · Authors · 2025-07-31
>
> We would like to thank you for your comprehensive review. We have carefully considered your comments and suggestions, and the following are our detailed responses.
>
>
> > **Question #1: How sensitive is CMGNN to the choice or accuracy of pseudo labels used in compatibility matrix estimation?**
>
> Answer #1:
> To investigate how sensitive CM estimation is to the accuracy of pseudo labels, we conduct an experiment to compare the quality of estimated CMs with different accuracy of pseudo labels.
> The results show that **the CM estimation is quite robust to the accuracy of pseudo labels.** This may be thanks to the confidence-based CM estimation.
>
> The details of the experiment are as follows:
> * For CM, we use the corresponding **CMD** to better measure its quality since the matrix is hard to visualize.
> * We use an additional low label setting (5% nodes for training) to construct different quality of pseudo labels.
> * We report the CMD and accuracy of pseudo labels (pse_acc) after training on the normal setting and low label rate setting, along with the CMD of the fixed CM in the original data.
>
> The results are as follows:
> |Dataset|Rom.|Rom.|Ama.|Ama.|Cha.|Cha.|Pub.|Pub.|
> |:-:|:-:|:-:|:-:|:-:|:-:|:-:|:-:|:-:|
> |**Setting**|**CMD**|**pse_acc**|**CMD**|**pse_acc**|**CMD**|**pse_acc**|**CMD**|**pse_acc**|
> |Fixed|0.847|-|0.458|-|0.274|-|1.396|-|
> |Normal|0.859|0.919|0.626|0.746|0.488|0.682|1.507|0.938|
> |Low Label Rate|0.834|0.648|0.592|0.416|0.417|0.397|1.490|0.830|
>
> From the above results, we can see that **CMGNN can estimate CMs with relatively stable CMD values**, even when the pseudo labels are of low quality in the low label rate setting. This demonstrates the robustness of CMGNN to the quality of pseudo labels.
>
>
> > **Question #2 & Weakness #2: In evolving graphs, would re-estimating CMD and retraining CMGNN be feasible since it introduces extra computation?**
>
> Answer #2: Thank you for this question! **CMGNN is largely adapted to the evolving graphs while introducing little computing consumption compared with normal graphs.** The detailed analyses are as follows:
> * For a newly arrived node, it will influence the following part of CMGNN: the **node itself**, the corresponding **class prototype**, and the estimated **compatibility matrix**.
> * For the increase of the node itself, there is **unavoidable time consumption** for all methods.
> * For new nodes with available labels, the corresponding class prototype in CMGNN should be updated too. And the **update is quite simple** if we record the number of nodes for each class, as the class prototype is set as the average of all nodes in the same class.
> * For the estimated CM, **the new node can be easily integrated into the update**, since the CM needs to be gradually updated during the training process. Of course, this will bring a little extra computational cost, as the number of updates may increase.
>
> In summary, **CMGNN can be easily adapted to the evolving graphs since the arrival of new nodes and labels meets the update strategy of CMGNN and brings little extra computation**.
>
>
> > **Question #3 and Weakness #1: Does CMD offer a reliable signal under label imbalance, since the theoretical results rely heavily on the CSBM-H model?**
>
> Answer #3:
> We investigate the effectiveness of CMD under a label imbalance setting theoretically and empirically as follows:
> * Theoretical analysis: The CMD is designed to measure the discriminability of CM. Since each row of CM is calculated for the corresponding class separately, the label imbalance will not impact the CM. Further, during the calculation of CMD, the contribution of all classes is the same, regardless of the node number of each class. Thus, **CMD can hold the effectiveness in the real-world label imbalance setting theoretically.**
> * Empirical analysis: We conduct experiments on 4 real-world datasets with label imbalance to evaluate the effectiveness of CMD. **The experimental results show that CMD can still offer a reliable signal under label imbalance.** Meanwhile, CMGNN achieves outstanding performance on these datasets, as shown in Table 1, also providing empirical proofs.
>
>
> The details of the experiment are as follows:
>
> To evaluate the effectiveness of CMD, we try to calculate the ENKL (node discriminability) and CMD on a real-world label imbalance setting, and investigate the relationship between them.
> The following are the experimental settings:
> * **CMD**: The CMDs of fixed CM and Weighted-CM on real datasets are easy to calculate according to the definition.
> * **ENKL**: The ENKL is designed to measure the relationship between two distributions, which corresponds to the two-class setting. To approximately estimate the ENKL on real datasets with multiple classes, we first group the node embeddings according to node labels, regarding them as observed samples of different class distributions. Further, we calculate the pair-wise ENKL among classes and report the weighted mean values as results.
> * **Settings**: For CMD, we report the fixed CM of the original data and the Weighted-CM of different operators. For ENKL, we report results of node embeddings after message passing (without feature transformation). We conduct the comparison of 4 operators (LP, HP, HTMP_0, HTMP_1) on 4 real-world datasets:
> $$
> \mathbf{H}^{LP}=\hat{\mathbf{A}}\mathbf{X},\quad
> \mathbf{H}^{HP}=(\mathbf{I}-\hat{\mathbf{A}})\mathbf{X},\quad
> \mathbf{H}^{HTMP_0}=(\mathbf{I}-\hat{\mathbf{A}}+\hat{\mathbf{A}}^2)\mathbf{X},\quad
> \mathbf{H}^{HTMP_1}=(\mathbf{I}+\hat{\mathbf{A}}+\hat{\mathbf{A}}^2)\mathbf{X}.
> $$
> To show the details of label imbalance in real-world datasets, we report the rate of node numbers of the largest and smallest classes and the proportion of classes:
> |Dataset|Amazon-Ratings|Roman-Empire|Squirrel-F|Actor|
> |:-:|:-:|:-:|:-:|:-:|
> |num_class|5|18|5|5|
> |Rate of max/min class|8.5|10|3.2|2.3|
> |class proportion (%)|(36.8, 26.8, 23.2, 8.9, 4.3)|(14.0, 13.8, 11.0, 11.0, 9.7, 6.0, 5.5, 4.8, 4.2, 3.8, 3.5, 3.2, 2.0, 1.9, 1.6, 1.5, 1.4, 1.4)|(34.0, 23.2, 17.9, 13.4, 10.5)|(25.9, 23.9, 21.4, 17.5, 11.2)|
>
> The results are as follows:
> |Dataset|Ama.|Ama.|Rom.|Rom.|Squ.|Squ.|Act.|Act.|
> |-|-|-|-|-|-|-|-|-|
> |**MP Operator**|**CMD**|**ENKL**|**CMD**|**ENKL**|**CMD**|**ENKL**|**CMD**|**ENKL**|
> |**LP**|0.46|-0.26|0.85|-0.76|0.08|-14.84|0.03|-2.77|
> |**HP**|1.95|-0.80|2.77|-2.62|2.02|-67.89|2.01|-9.28|
> |**HTMP_0**|2.35|-1.14|3.31|-4.13|2.22|-80.05|2.29|-11.53|
> |**HTMP_1**|3.10|-2.41|3.55|-5.04|2.28|-89.52|2.31|-11.63|
>
> In line with expectations, the CMD and ENKL show **an obvious negative correlation on all test datasets**. This illustrates that Theorem 3.2 still holds empirically on real-world datasets with label imbalance, indicating the effectiveness of CMD.
>
>
> **We are expecting these could help answer your questions and look forward to further discussion. If you have any further questions, please feel free to contact us.**

---

> > ### Comment · Reviewer_5t7f · 2025-08-06
> > **Replying to Rebuttal**
> >
> > Thank you for the reply and comprehensive experiments added. Most of my main concerns have been well addressed and the authors' effort during the rebuttal phase has made this work much more solid, I will increase my **confidence score from 3 to 4**.

---

> > > ### Author Response · Authors · 2025-08-07
> > >
> > > Thank you for your positive feedback and for raising your confidence score! We sincerely appreciate the time and care you invested in reviewing our work, which has significantly enhanced its quality. We will incorporate these analyses into the revised version.

---

### Note · Authors · 2025-08-12

We are sincerely grateful to the Area Chairs and all reviewers for their time and effort in reviewing our paper, as well as for **the recognition of our contributions**.
We are pleased to see that our work has been recognized by the reviewers in **well-motivated theoretical analysis and methodology** (Reviewers 5t7f, ZGc9), **comprehensive experimental evaluation** (Reviewers 5t7f, tib1, eeXf, ZGc9), **superior experimental performance** (Reviewers 5t7f, tib1, eeXf, SFcH, ZGc9), and **logically-structured writing** (Reviewers 5t7f, ZGc9).
During the rebuttal and discussion phase, we successfully **addressed the majority of the reviewers' concerns** and received further **positive feedback**, as evidenced by:
* Reviewer **tib1** raised the score to positive.
* Reviewer **5t7f** increased the confidence score of the high rating.
* Reviewer **ZGc9** maintained the positive score.

We also appreciate the opportunity to engage in **further discussion** with Reviewer **eeXf**. While we note there has been no additional response to our clarifications, we remain hopeful that Reviewer eeXf finds the reply satisfactory and would be grateful if reconsidering the final rating. We remain fully prepared to address any further questions.

To summarize our rebuttal efforts, we clarified and supplemented our work in the following key areas:

* **Empirical Validation of Theorem 3.2 on Real-World Graphs**: We demonstrated the effectiveness of Theorem 3.2 on real-world graphs that beyond theoretical assumptions, addressing the concerns about strong assumptions.
* **Discussion of Theoretical Contributions**: We contrasted our work with existing works, explicitly highlighting our novel contributions.
* **Enhanced Experimental Evaluation**: We incorporated GraphSAGE as an additional baseline and conducted more comprehensive ablation studies, quantitatively validating the contribution of each component.
* **Robustness Verification**: We analyzed CMGNN's performance on heterophilic graphs and under low label rate settings, reconfirming its effectiveness and robustness.
* **Point-by-Point Clarifications**: We addressed the detailed questions raised by the reviewers comprehensively.

Once again, we deeply appreciate everyone's efforts and respect diverse perspectives.

---

### Decision · Program_Chairs · 2025-09-17

**Decision:**

Accept (poster)

**Comment:**

This paper introduces CMGNN, which leverages a new structural–label metric, Compatibility Matrix Discriminability (CMD), to improve performance on heterophilous graphs through confidence-based CM estimation, virtual neighbor supplementation for low-degree nodes, and CM-aware aggregation. Experiments on 13 datasets with 19 baselines show consistent gains, especially under low degree or label rates, supported by robustness tests, ablations, and a unified codebase. While theoretical analysis is limited to simplified settings and some design choices seem engineered, the empirical breadth, clear motivation, and practical utility make the contribution strong. I recommend accept, with camera-ready improvements including clarifying theoretical scope, tightening exposition, and contextualizing the benefits and overhead of individual components.